# TOOLWEAVE: FINE-GRAINED AND CONTROLLABLE SYNTHETIC DATA GENERATION FOR MULTI-TURN TOOL CALLING WITH NON-FRONTIER LLMS

## ABSTRACT

Multi-turn tool-calling is a crucial capability for LLM-based agents and is typically improved via supervised fine-tuning on synthetic data. Existing multi-turn tool-calling synthetic data pipelines often rely on proprietary frontier LLMs (e.g., GPT-4) or commercial APIs (e.g., RapidAPI), introducing restrictive licensing. In contrast, data generated directly from non-frontier LLMs suffers from low fidelity, poor diversity, and weak adherence to multi-constraint instructions, resulting in producing lower-quality datasets than frontier models. To address these limitations, we propose *ToolWeave*, a modular and controllable pipeline that synthesises high-quality multi-turn tool-calling datasets using non-frontier, license-friendly LLMs. ToolWeave supports both API and dialogue synthesis. Our framework's novelty is threefold: (1) it is fully synthetic; given only a domain name, it builds a domain context from Wikipedia and Wikidata to synthesize a *Tool Graph* of APIs. (2) In contrast to other pipelines' single, failure-prone planning step, Tool-Weave's *scaffolding* process first generates a high-level goal from the Tool Graph, then decomposes it into a turn-level dialogue plan. This two-stage approach enables non-frontier LLMs to generate high-fidelity, grounded dialogues. (3) A final post-processing stage injects lexical diversity and robustness patterns (e.g., error recovery) to simulate real-world scenarios. To validate our framework, we generated a dataset of ~3.2k dialogues using the open-source `gpt-oss-120b`. Compared to baselines, ToolFlow and ToolDial, ToolWeave shows clear gains: on the BFCL-V3 benchmark, our data improves Llama-3.1-70B to **33.25%** (vs. ToolFlow's 21.00% & ToolDial's 3.75%) and Phi-4 to **24.50%** (vs. ToolFlow's 8.88% & ToolDial's 2.0%). Our data also shows strong generalization, with peak gains of **37.6%** on the API Bank benchmark.

## 1 INTRODUCTION

The ability to leverage external tools has transformed large language models (LLMs) from static predictors into autonomous agents capable of accomplishing complex, real-world tasks(Yao et al., 2023; Schick et al., 2023; Paranjape et al., 2023). A central capability for such agents is *multi-turn tool calling*: identifying the right tool, chaining multiple tools to complete a workflow, eliciting missing information from the user, and presenting results back. A common strategy to improve this ability is fine-tuning on synthetic datasets (Liu et al., 2025; Prabhakar et al., 2025; Shim et al., 2025; Qin et al., 2024). However, existing pipelines were designed for frontier LLMs and make assumptions that break down when applied to non-frontier, open models.

We identify four critical failure modes in prior work. (1) To obtain multi-turn, multi-tool workflows, many pipelines build a graph of tools and perform a random walk over it. This often produces incoherent tool sequences for dialogues, leading to *less realistic data*. (2) Given such tool sequences, pipelines usually ask an LLM to generate the entire dialogue plan in one shot. Frontier LLMs can manage this, but weaker LLMs struggle with such broad prompts, causing *complex instruction following* failures. (3) *Dialogue drift and state loss*: even when a plan exists, generating the dialogue directly from a high-level plan often leads to hallucinated arguments, loss of parameter provenance, or contradictions across turns. (4) *Licensing and coverage constraints*: many pipelines rely on APIs from marketplaces like RapidAPI, causing licensing issues and limited domain coverage.

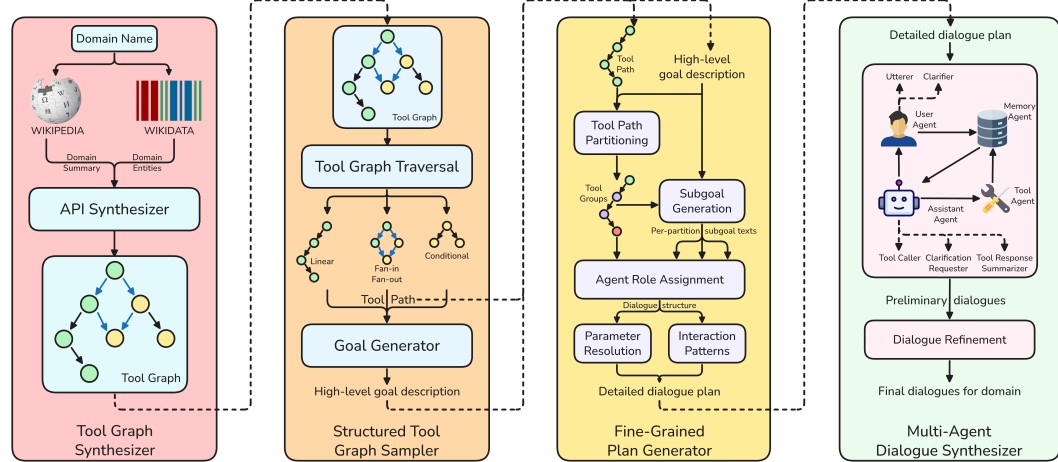

Figure 1: The modular architecture of ToolWeave. Starting with a domain name, the *Tool Graph Synthesizer* creates synthetic APIs and organizes them into an interconnected graph; the *Structured Tool Graph Sampler* extracts coherent subgraphs as goals; the *Fine-Grained Plan Generator* expands these goals into detailed plans; and the *Multi-Agent Dialogue Synthesizer* instantiates the plans into natural multi-turn conversations.

**Our approach:** We introduce *ToolWeave*, a controllable pipeline that synthesizes domain-specific tools and realistic multi-turn tool-calling dialogues using only open knowledge sources and models (e.g., gpt-oss-120b). Given a domain label (e.g., customer_service), ToolWeave (1) synthesizes a license-friendly library of tools and organizes them into a Tool Graph (where nodes represent APIs and edges represent validated parameter dependencies) , (2) samples coherent subgraphs from structural workflow motifs, (3) produces explicit, fine-grained plans that decomposes dialogue planning into deterministic atomic steps (specifying agent roles, subgoals, and parameter sources at every turn), and (4) executes them via a plan-driven multi-agent synthesizer with dynamic roles and a Memory agent to preserve state, followed by post-processing for diversity and robustness (Figure 1).

**Design principle: granularity and control.** Prior pipelines rely on heuristics to construct edges in the tool graph, high-level goal planning, and one-shot dialogue synthesis from coarse plans. In contrast, ToolWeave introduces validated, fine-grained control across all stages of the data synthesis process. Each component—API synthesis, tool-graph construction, workflow sampling, subgoal planning, and multi-agent execution—produces controlled, verifiable inputs for the next stage, enabling non-frontier LLMs to reliably generate complex multi-turn tool-calling dialogues. This end-to-end scaffolding enforces consistent data flows and small, verifiable reasoning steps, providing structural reliability that prior pipelines lack.

The key novelties of ToolWeave and how they map to failure modes are:

- *Knowledge-grounded synthetic tools* address licensing and coverage constraints by generating tools from open textual and structured sources, producing domain-aware schemas without proprietary APIs.
- *Structured tool-graph sampler* addresses diversity/realism by sampling workflows from structural motifs (linear, fan-in/out, conditional) rather than random walks.
- *Fine-grained plan generator* addresses complex instruction following by decomposing planning into targeted steps that non-frontier LLMs can reliably execute.
- *Plan-driven multi-agent synthesizer with Memory* addresses drift and state loss by executing plans via dynamic-role agents and explicit state tracking.

LLMs fine-tuned on ToolWeave data generated using gpt-oss-120b outperform SOTA baselines *ToolFlow* (Wang et al., 2025) and *ToolDial* (Shim et al., 2025) across multiple benchmarks, including BFCL-V3 (Patil et al., 2025a), API Bank (Li et al., 2023), CONFETTI (Alkhouli et al., 2025), and ToolHop (Ye et al., 2025). We will release code, synthesized tools, and dialogues upon acceptance.

## 2 RELATED WORK

**Fine-Tuning Datasets and Models for Tool Calling:** Toolformer (Schick et al., 2023), one of the first works, replaced text segments with tool calls to train LLMs to produce tool calls. Several instruction-tuning datasets have been created to improve tool-calling capabilities in LLMs. Most of these datasets are designed to introduce single-turn tool-calling capabilities in models (Liu et al., 2024; Basu et al., 2024a;b; Shi et al., 2025; Qin et al., 2024) using APIs extracted from sources such as RapidAPI (RapidAPI, 2025) or APIBench (Patil et al., 2025b). Reinforcement learning methods and other tuning strategies to improve tool calling capabilities are presented in Li et al. (2025); Chen et al. (2025b). Tool retrieval becomes an important aspect with increasing tools and is addressed in the works Qin et al. (2025); Chen et al. (2024). Recently, researchers have started looking at creating multi-turn tool calling data using synthetic data generation pipelines and models (Wang et al., 2025; Shim et al., 2025; Prabhakar et al., 2025; Liu et al., 2025; Yin et al., 2025; Chen et al., 2025a). The APIs and frontier models used in these pipelines make them very restrictive and not licensable for commercial deployments. The goal of ToolWeave is to address this gap with an end-to-end license-friendly open synthetic data pipeline that gives control at each stage without any restrictions. Table 1 provides the comparison of the features around licenses of the model used, the code, the generated data and the fine-grained control it offers.

**Tool Calling Evaluation Benchmarks:** BFCL (Patil et al., 2025a) is one of the comprehensive benchmarks aimed at evaluating various aspects of function calling including multi-turn. Apart from this, several other benchmarks evaluate single-turn capabilities of the models like (Wu et al., 2024; Patil et al., 2025a; Ross et al., 2025; Shi et al., 2025; Xu et al., 2023; Zhuang et al., 2023; Huang et al., 2024; Basu et al., 2024a). Benchmarks like CONFETTI (Alkhouli et al., 2025), APIBank (Li et al., 2023), ToolHop Ye et al. (2025) are built for evaluating multi-turn capabilities of the models. $\tau$-bench (Yao et al., 2025) and Agentboard (Chang et al., 2024) are benchmarks for evaluating agentic capabilities such as multi-turn tool calling along with policy adherence.

Table 1: Comparison of multi-turn synthetic data pipelines across different dimensions.

| Framework | Synthesize APIs | Code Released | Data Released, License Friendly | License Friendly Synthesis Model | Fine-Grained Plan |
|---|---|---|---|---|---|
| Tooldial (Shim et al.) | ✗ | ✓ | ✓, ✗ | ✗ | ✗ |
| ToolAce (Liu et al.) | ✓ | ✗ | ✓, ✗ | ✗ | ✗ |
| APIgen-MT (Prabhakar et al.) | ✗ | ✗ | ✓, ✗ | ✗ | ✗ |
| ToolFlow (Wang et al.) | ✗ | ✗ | ✗, ✗ | ✗ | ✗ |
| Button (Chen et al.) | ✓ | ✗ | ✓, ✗ | ✗ | ✗ |
| ToolLLM (Qin et al.) | ✗ | ✓ | ✓, ✗ | ✗ | ✗ |
| ToolWeave (Ours) | ✓ | ✓* | ✓,*✓ | ✓ | ✓ |

\* Will be released upon paper acceptance.

## 3 TOOLWEAVE ARCHITECTURE

The architecture of ToolWeave is guided by a core design philosophy: to enable non-frontier LLMs to generate high-quality, complex, license-friendly tool-calling data. To achieve this, we designed a *modular, fine-grained framework* that provides precise "scaffolding" at each stage as shown in Figure 1. The process begins with the *Tool Graph Synthesizer*, which creates a pool of domain-specific synthetic APIs (or tools) and organizes them into an interconnected *Tool Graph*. Next, the *Structured Tool Graph Sampler* searches the Tool Graph to find coherent subgraphs that represent plausible goals. These goals are then expanded by the *Fine-Grained Plan Generator*, which decomposes them into subgoals and generates a fine-grained plan. Next, the *Multi-Agent Dialog Synthesizer* instantiates this plan into a natural, multi-turn conversation. Finally, the *Dialogue Post-Processor* refines the dialogue, injecting realistic error patterns and lexical diversity to create a robust final dataset. Appendix I.1 to I.4 demonstrate the output of each stage through an example.

### 3.1 TOOL GRAPH SYNTHESIZER

**Motivation:** Prior work in multi-turn tool-calling dialogues, such as ToolDial (Shim et al., 2025) and ToolFlow (Wang et al., 2025) has largely relied on real-world APIs from marketplaces such as

RapidAPI (RapidAPI, 2025). However, such resources are subject to licensing and usage restrictions that preclude their use in commercial training (RapidAPI, Inc., 2025). Moreover, for academic research they hinder reproducibility (Guo et al., 2024) and experimental control, as APIs are volatile and their fixed schemas limit systematic studies of properties such as schema complexity. In addition, these marketplaces have skewed and limited coverage, often lacking APIs for new or specialized domains. To address these challenges, our *ToolWeave* framework employs *synthetic tool generation*, a scalable, safe, and fully controllable alternative that ensures *breadth* (many distinct tools), *depth* (realistic schemas), and *connectivity* (linked and composable tools).

**Key idea:** Our key insight is to bootstrap the process from openly available domain knowledge. Starting with a domain string (e.g., `customer_support`), we retrieve two complementary views of the domain: (1) narrative summaries from Wikipedia (Wikipedia, 2025) and (2) structured entities from Wikidata (Wikidata, 2025). This grounding step provides domain-specific vocabulary, roles, and objects that seed tool schemas with realistic names and argument structures. For instance, in the `travel_booking` domain, Wikipedia may introduce workflows around itineraries, while Wikidata supplies entities such as airports, cities, and airlines, together yielding realistic tool definitions.

**Iterative, curriculum-driven synthesis:** A tempting solution would be to simply ask an LLM to generate all the APIs for a domain. In practice, this produces only obvious functions (e.g., in `travel`, `search_flight` or `book_flight`), missing broader workflows such as loyalty or reviews. Moreover, LLMs invent parameters with conflicting types and naming conventions. The resultant tools rarely share arguments that enable the composition of tools. Hence, rather than asking an LLM to generate all the tools in one step, we follow a *synthesis plan*—a curriculum that decomposes the task into progressively harder stages.

1. *Seed generation:* Generate a minimal set of core APIs covering fundamental workflows.
2. *Entity expansion:* Integrate APIs tied to Wikidata entities, broadening coverage.
3. *Schema enrichment:* Add depth with nested objects, enums, defaults, and optional parameters.
4. *Connection discovery and Tool Graph Construction:* In contrast to prior work (Shim et al., 2025; Wang et al., 2025), where edges of the tool graph are inferred using semantic similarity, ToolWeave generates tools with candidate links in mind (e.g., `search_hotel` API produces a `hotel_id` specifically for `book_hotel` API to consume). These proposed edges are then validated by an LLM to confirm that data flows are semantically correct.

After each stage, candidate APIs are syntactically validated, deduplicated with embedding-based checks, and refined. This incremental approach ensures APIs become richer and more connected over time. The output of this stage is a domain-specific Tool Graph $\mathcal{G}$ and an API pool $\mathcal{A}$ that are diverse, interconnected, and semantically robust.

## 3.2 Structured Tool Graph Sampler

**Motivation:** Once the Tool Graph is constructed, we identify subgraphs that can be used to construct user goals. A common strategy, used in ToolFlow (Wang et al., 2025), is to perform a random walk on the Tool Graph to get a tool sequence. This ensures that tools are *syntactically compatible*—the output of one tool can be used as the input of the next—but it does not guarantee that the tools form a *coherent goal*. For instance, a tool that returns a `user_id` could be randomly chained to a tool that deletes an account, even if the more natural follow-up would be to query a profile or update preferences.

**Key idea:** Instead of assuming that type-compatible tools naturally form a workflow, ToolWeave enforces *tool-goal alignment* by identifying subgraphs that represent coherent user objectives. To do this, sampling is restricted to structural motifs (linear, fan-in/fan-out, conditional) that reflect common task patterns, followed by validation for semantic plausibility. This ensures that every selected path corresponds to a realistic, goal-directed sequence of tools.

**Process:** The Structured Tool Graph Sampler operates in three stages.

- *Tool-Graph traversal:* Subgraphs are extracted by searching the Tool Graph for common workflow motifs. We focus on three structural patterns: (1) *linear paths*, discovered through bounded beam search and representing sequential workflows such as `search_flight` → `book_flight`, (2)

*fan-in/fan-out patterns*, identified by tracing nodes that feed into multiple successors or aggregate multiple inputs, capturing parallel subtasks such as `get_user_profile` branching into both `search_flights` and `search_hotels`, which later converge on `book_package`, and (3) *conditional branches*, extracted by scanning output schemas for boolean or enum fields, introducing decision-making, e.g., calling `retry_payment` if `payment_success=false`.

- *Path Ranking and Filtering:* Candidate paths/subgraphs are scored and filtered before conversion to natural-language goals. For linear paths, we retain the top-$k$ paths by applying *Maximal Marginal Relevance (MMR)* (Carbonell & Stewart, 1999). MMR operates on semantic embeddings of the natural language goal descriptions for each path, ensuring a balance between path relevance and diversity. Fan-in/fan-out patterns are constrained by schema-compatibility, while conditional candidates are validated by checking that the predicate field exists and that downstream tools consume the relevant output type. The detailed implementation is described in Appendix C. For a visualization of a representative tool graph illustrating the specific structural patterns discussed (linear, fan-in-fan-out, and conditional), please refer to Appendix B.3.
- *Goal description generation:* Each retained tool path is then converted into a natural-language goal using an LLM. We prompt the LLM with the full tool sequence, the respective API schemas, and the specific pattern type (e.g., linear, fan-in/out). For conditional patterns, the prompt also includes the branch predicates to ensure the final goal contains the necessary branching logic.

The output of this stage is a structured set of goal objects, each containing the tool path, motif type, and a goal description, which serve as inputs to the Fine-Grained Plan Generator.

## 3.3 FINE-GRAINED PLAN GENERATOR

**Motivation:** Given a candidate goal (a natural-language objective and its corresponding tool path), the challenge is to map this abstract sequence of tool calls into a *coherent, multi-turn conversation*. A naïve approach would be to ask an LLM to directly generate the dialogue end-to-end. However, this is fragile, as the model may hallucinate parameter values, confuse which arguments come from the user vs. prior tool outputs, or fail to insert necessary clarification turns for missing information. While frontier models such as GPT-5 OpenAI (2025) are less prone to such mistakes, open-source non-frontier LLMs often struggle with this mapping, producing ambiguous or incoherent dialogues.

**Key idea:** To prevent such errors, ToolWeave *explicitly controls* this mapping through the *Fine-Grained Plan Generator:* Rather than relying on an LLM's implicit reasoning, we deterministically construct a step-by-step JSON plan that encodes: (1) which agent acts at each diaogue step, (2) which subgoal is advanced, (3) which tool(s) must be invoked, (4) how parameters are sourced (user-supplied, upstream tool outputs, or schema defaults), and (5) where clarifications should be injected. This plan serves as precise scaffolding that non-frontier LLMs can follow without drifting.

**Process:** The Fine-Grained Plan Generator proceeds in three steps:

1. *Partitioning the tool path:* The input tool sequence is partitioned into turn-level groups (e.g., `[A,B,C,D] → [[A],[B,C],[D]]`). For linear paths, this is done by an LLM conditioned on the goal text to ensure natural subgoal boundaries; for fan-in/fan-out or conditional patterns, deterministic rules preserve structural dependencies.
2. *Subgoal synthesis and parameter planning:* Each partition is associated with a concise subgoal utterance (e.g., "First, retrieve the user's account details"). Then, for each tool in the partition, the planner resolves parameters by tracing the Tool Graph: *user-supplied* parameters are explicitly requested in a user turn, *clarification-gated* parameters are withheld to trigger a clarification turn, and *derivable* parameters are filled automatically from previous tool outputs or schema defaults. This ensures a precise data flow with no ambiguity.
3. *Plan construction:* The planner encodes the dialogue as an ordered sequence of JSON steps. Each step specifies the `role` (`user`, `assistant`, or `tool`), the `type` of action (`USER_UTTERANCE`, `TOOL_OUTPUT`, `ASSISTANT_CLARIFICATION`, etc.), the subgoal (if any), and all parameter bindings with explicit source hints. The planner also inserts micro-interactions, such as clarification turns (`ASSISTANT_CLARIFICATION → USER_RESPONSE`) or optional chit-chat, making conversations natural while preserving determinism.

The output is an ordered, fine-grained JSON plan that clearly specifies the dialogue structure.

### 3.4 MULTI-AGENT DIALOGUE SYNTHESIZER

**Motivation:** Once a fine-grained JSON plan has been generated, the challenge is to *instantiate* it into a coherent, natural multi-turn dialogue. A straightforward option would be to let an LLM read the plan and generate the conversation freely. However, this creates two problems, particularly for non-frontier LLMs: (1) they often lose track of long-range state (e.g., which parameter values were resolved earlier), and (2) they may deviate from the plan, skipping steps or hallucinating tool calls. Thus, while the plan provides structure, a mechanism is needed to enforce faithful execution.

**Key idea:** ToolWeave addresses this with a *plan-driven multi-agent execution framework*, where specialized agents follow the plan under strict role control. The `User` and `Assistant` act as *dynamic-role agents*, adopting different personas depending on the step (e.g., Utterer vs. Clarifier for the User, Tool Caller vs. Summarizer for the Assistant). A dedicated *Memory agent* maintains an explicit JSON state of facts, resolved parameters, and tool outputs, offloading context tracking from the LLM. Finally, a *Tool Simulator* enforces schema-consistent outputs for tool calls, ensuring plausibility without live APIs. This design keeps execution faithful to the plan while maintaining natural dialogue flow.

**Process:** The synthesis process is a deterministic, stateful loop that executes the JSON plan step-by-step. At each step, the plan dictates the agent's `role`; that agent is then invoked with the current dialogue history and Memory state. The agent's output (e.g., an utterance or tool call) is appended to the transcript and used to update the Memory before proceeding to the next step. This stage yields coherent, plan-aligned, and tool-grounded dialogues, which are then refined in the Dialogue Post-Processing stage.

### 3.5 DIALOGUE POST-PROCESSING

The final stage of the ToolWeave framework, *Dialogue Post-Processing*, takes the "clean" dialogues from the Synthesizer and transforms them into a robust, challenging, and realistic final dataset as described below.

1. *Improving Naturalness and Diversity:* To address the tendency of non-frontier models to produce repetitive utterances, we apply *User Utterance Paraphrasing*. An LLM paraphrases user turns, increasing lexical and syntactic variety while preserving the turn's subgoal and slot values. This reduces the risk of overfitting to repeated user phrasing.
2. *Injecting Robustness and Refusal Logic:* We further enhance robustness by deterministically injecting real-world challenges, including *Error Recovery* (introducing erroneous tool calls with realistic failures) and *Missing-Function* scenarios (temporarily withholding an API to train calibrated refusal and recovery once the schema is reintroduced). Detailed algorithms for these error injection strategies are presented in Appendix F.
3. *Preventing Shortcut Learning:* To prevent shortcut learning and force the model to reason over schemas instead of memorizing names, we apply *Masking*, which systematically replaces API and argument names with generic IDs (e.g., `func_01`, `arg_01`).

Together, these transformations produce a final dataset that is diverse, robust, and challenging.

## 4 ANALYSIS OF DATA QUALITY

### 4.1 API QUALITY AND COVERAGE

We evaluate our synthetic APIs on three key properties. First, we measure *breadth* using the average number of APIs per domain and input parameters per API. Second, we assess *depth* (schema complexity) using two key metrics: the *Complex API Use (CAU)*—the proportion of APIs with nested objects or arrays—and the *Required Parameter Ratio (RPR)*, the proportion of an API's parameters that are required. Finally, we evaluate *connectivity* using *Interconnectivity (IC)*, which measures direct data-flow potential, and the average longest tool chain. (Formal definitions for all metrics are in Appendix B.7). We generated API sets for 20 domains ( B.2) using the `gpt-oss-120b` and `mistral-medium-2505` as synthesis LLMs. Per-domain averages, shown in Table 2a, reveal an interesting trade-off: the open-source model produces APIs with greater *breadth* and *depth*, while

Table 2: Comparison of API statistics (left) and dialogue statistics (right).

(a) Avg. API stats per domain.

| Metric | GPT-OSS | Mistral |
|---|---|---|
| Avg. APIs / Domain | 25.25 | 20.85 |
| Avg. Params / API | 3.60 | 2.92 |
| Complex API Use (CAU) | 23.6% | 19.8% |
| Required Param Ratio (RPR) | 65.6% | 68.4% |
| Interconnectivity (IC) | 1.08 | 1.85 |
| Longest Chain (avg.) | 5.5 | 5.9 |

(b) Statistics of synthetic dialogues.

| Dataset | Min/Max/Avg. Turns | Min/Max/Avg. Tool Calls | % Total/ % True Multi-step Turns |
|---|---|---|---|
| ToolWeave (GPT-OSS) | 1/8/2.96 | 1/6/3.24 | 35.69/31.69 |
| ToolWeave (Mistral) | 1/7/2.49 | 1/5/3.20 | 44.12/36.14 |
| ToolFlow (GPT-OSS) | 1/69/5.65 | 0/37/2.45 | 4.45/0.0 |
| ToolDial (GPT-4o) | 2/6/4.48 | 0/2/1.51 | 4.47/0.0 |

the proprietary model excels at creating a more tightly interconnected *Tool Graph*. This shows our framework guides both models to generate high-quality, but stylistically different APIs.

**Comparison with Real-World APIs:** We compare our synthetic APIs to 4,474 real-world APIs from RapidAPI (used in ToolDial (Shim et al., 2025)). Our APIs show higher complexity, with an average parameter count of 3.6 vs. 2.4 and a *Complex API Use (CAU)* score of up to 23.6% (vs. 1.0% for real APIs). In connectivity, our method achieves an Interconnectivity score of up to 1.85. For ToolDial, we compute the Interconnectivity score in two ways: *exact parameter name matching* yields 1.61, while *semantic similarity* based on parameter descriptions (threshold 0.82) raises it to 2.37. Both inflate scores with false positives: exact matching links generic names (e.g., id), while semantic similarity provides only a "soft" match. By contrast, our generative approach with LLM validation prunes such edges, producing a cleaner, causally-sound *Tool Graph*.

## 4.2 DIALOGUE STRUCTURE AND COMPLEXITY

Using the APIs generated for 20 domains, we synthesize two distinct datasets of ∼3.2k dialogues each with ToolWeave (using gpt-oss-120b and mistral-medium-2505 as synthesis LLMs). For a direct comparison, we also generate a comparable ∼3.2k dialogue baseline using the *ToolFlow* pipeline, driven by the same gpt-oss-120b model. We compare their structure in Table 2b. We measure the number of *turns* (interactions between user utterances) and the total *tool calls* per dialogue. Crucially, we also measure the percentage of *multi-step turns*. A turn is considered multi-step if it requires at least two tool calls. We further define a *True* multi-step turn as one where these tool calls are directly dependent, meaning the output of one tool call is consumed by another within the same turn. For example, a turn where an assistant calls search_flights and immediately uses the resulting flight_id to call book_flight is a *True* multi-step turn. The results highlight a key difference in generation quality. The unusually high maximum turn and tool call counts for ToolFlow (69 and 37, respectively) do not indicate complexity, but rather uncontrolled generation; this is evidenced by its *0.0% True multi-step turn rate*. Without a fine-grained plan, the non-frontier LLM enters long, unproductive loops of shallow, independent tool calls. In contrast, ToolWeave's detailed plans produce shorter, more focused dialogues that are dense with genuine multi-step reasoning (up to *36.14% True multi-step*), demonstrating a more effective and controlled synthesis process. ToolDial (Shim et al., 2025), despite leveraging GPT-4o with real APIs, yields shallow dialogues averaging only ∼1.5 tool calls in a conversation and no true multi-step turns. Some examples of these issues for ToolFlow and ToolDial are provided in Appendix J and K.

## 4.3 LLM-AS-JUDGE EVALUATION OF DIALOGUES

Following ToolFlow (Wang et al., 2025), we adopt an LLM-as-judge evaluation protocol to assess the semantic quality of dialogues. Specifically, we employ Llama-3-405B-Instruct (Grattafiori et al., 2024) to evaluate a random sample of 200 dialogues from each dataset, distributed evenly across all 20 domains (except for ToolDial). The judge rates each dialogue on a scale of 1-5 along four dimensions: *Naturalness* (human-likeness of conversation), *Coherence* (logical flow and relevance),

Table 3: LLM evaluation of synthetic dialogues. Nat=Naturalness, Coh=Coherence, Hel=Helpfulness, Acc=Accuracy.

| Dataset | Nat | Coh | Hel | Acc |
|---|---|---|---|---|
| ToolWeave (GPT-OSS-seed) | 4.74 | 4.96 | 4.97 | 4.95 |
| ToolWeave (Mistral-seed) | 4.76 | 4.99 | 4.99 | 4.98 |
| ToolFlow (GPT-OSS-seed) | 4.51 | 4.83 | 4.87 | 4.79 |
| ToolDial (GPT-4o) | 4.09 | 4.68 | 4.53 | 4.91 |

*Helpfulness* (usefulness of the information provided), and *Accuracy* (factual correctness).

Table 3 summarizes the results. ToolWeave outperforms baselines across all four dimensions. Score differences across different synthesis seeds are $\leq 0.03$, indicating backbone-agnostic robustness. These findings complement our structural analysis in Table 2b. ToolFlow's comparatively lower ratings align with its uncontrolled, loop-heavy dialogues, which undermine coherence and helpfulness. ToolDial achieves relatively high accuracy but lags in naturalness and helpfulness. By contrast, ToolWeave's fine-grained planning, structural motifs, and post-processing yield dialogues that are not only structurally complex but also conversationally natural, coherent, and useful.

# 5 EXPERIMENTS

In this section, we experimentally evaluate the efficacy of our *ToolWeave* framework. Our goal is to answer three key questions: (1) Does fine-tuning on ToolWeave data substantially improve the multi-turn, multi-step tool-calling capabilities of base LLMs? (2) How does our data perform compared to prior state-of-the-art baselines, *ToolFlow* (Wang et al., 2025) and *ToolDial* (Shim et al., 2025)? (3) Do these performance gains generalize across different model families (Llama-3.1, Phi-4) and a diverse suite of evaluation benchmarks?

## 5.1 EXPERIMENTAL SETUP

**Training Datasets:** We experiment with four distinct training datasets. We generate two distinct datasets using ToolWeave framework, each containing ~3.2k dialogues: one seeded with the proprietary `mistral-medium-2505` model (Mistral AI, 2025), and one with the open-source `gpt-oss-120b` model (Agarwal et al., 2025). We compare these against two strong baselines. First, we use the full, publicly available ToolDial training dataset, consisting of 8859 dialogues synthesized with GPT-4o. Second, as the original implementation and data are not public, we re-implemented the ToolFlow pipeline and generated a comparable dataset of ~3.2k dialogues. For a fair comparison, both our ToolWeave and our re-implemented ToolFlow datasets were generated using the exact same `gpt-oss-120b` model and the same set of synthetic APIs from our API generation module (Section 3).

**Models and Fine-Tuning Setup:** We evaluate three base models to assess performance across different scales and architectures: *Llama-3.1-8B-Instruct*, its larger counterpart *Llama-3.1-70B-Instruct* (Grattafiori et al., 2024), and *Phi-4* (Abdin et al., 2024). All models are fine-tuned using the `LLaMA-Factory` framework[1] (Zheng et al., 2024) with `QLoRA` (Dettmers et al., 2023). Detailed hyperparameters are provided in Appendix G.

**Evaluation Benchmarks:** We evaluate all models on a diverse suite of four multi-turn tool-calling benchmarks: *BFCL-V3* (Patil et al., 2025a), a live, executable environment focused on complex conversational scenarios and error recovery; *API Bank* (Li et al., 2023) and *CONFETTI* (Alkhouli et al., 2025), which evaluate turn-level accuracy on fixed dialogue trajectories; and *ToolHop* (Ye et al., 2025), a multi-step benchmark with locally executable tools. Detailed descriptions of each benchmark are provided in Appendix H.

## 5.2 RESULTS AND ANALYSIS

### 5.2.1 PERFORMANCE ON BFCL-V3

Table 4 summarizes our results on the BFCL-V3 benchmark, which is designed to test complex conversational flows and robustness.

The results in Table 4 reveal four key trends:

1. **Cross-Family Efficacy:** ToolWeave-generated data transfers effectively across model architectures. For instance, the `Phi-4` model starts at **3.12%** but improves to **24.50%** after fine-tuning.
2. **Intra-Family Scaling:** Larger models within the same family benefit more from the `GPT-OSS-seed` data. `Llama-3.1-70B-Instruct` improves from **12.50%** to **33.25%**, outperforming its smaller 8B counterpart.

---

[1] https://github.com/hiyouga/LLaMA-Factory

Table 4: Comparison of model performance across BFCL-V3 multi-turn categories.

| Model | Multi Turn Acc | Multi Turn Base | Multi Turn Miss Func | Multi Turn Miss Param | Multi Turn Long Context |
|---|---|---|---|---|---|
| Llama-3.1-8B-Instruct | 9.25 | 11.00 | 8.00 | 8.50 | 9.50 |
| + FT on ToolDial (GPT-4o-seed) | 1.75 | 0.50 | 3.50 | 2.50 | 0.50 |
| + FT on ToolFlow (GPT-OSS-seed) | 7.62 | 11.00 | 4.50 | 5.50 | 9.50 |
| **+ FT on ToolWeave (Mistral-seed)** | **25.00** | **31.50** | 20.50 | **24.50** | **23.50** |
| **+ FT on ToolWeave (GPT-OSS-seed)** | 19.88 | 23.00 | **21.50** | 15.50 | 19.50 |
| Llama-3.1-70B-Instruct | 12.50 | 17.00 | 13.00 | 10.50 | 9.50 |
| + FT on ToolDial (GPT-4o-seed) | 3.75 | 4.50 | 3.00 | 4.00 | 3.50 |
| + FT on ToolFlow (GPT-OSS-seed) | 21.00 | 22.50 | 24.50 | 18.50 | 18.50 |
| **+ FT on ToolWeave (Mistral-seed)** | 28.88 | 27.50 | **39.00** | 30.00 | 19.00 |
| **+ FT on ToolWeave (GPT-OSS-seed)** | **33.25** | **37.50** | 35.00 | **32.00** | **28.50** |
| Phi-4 | 3.12 | 7.50 | 0.00 | 2.50 | 2.50 |
| + FT on ToolDial (GPT-4o-seed) | 2.00 | 2.00 | 0.50 | 3.50 | 2.00 |
| + FT on ToolFlow (GPT-OSS-seed) | 8.88 | 11.50 | 7.50 | 10.50 | 6.00 |
| **+ FT on ToolWeave (Mistral-seed)** | 19.37 | 20.50 | 23.50 | 20.50 | 13.00 |
| **+ FT on ToolWeave (GPT-OSS-seed)** | **24.50** | **24.00** | **28.50** | **31.00** | **14.50** |

3. **Targeted Capability Improvement:** Fine-tuning with ToolWeave data yields notable robustness on challenging categories. Base models often fail on *Missing Function* and *Missing Parameter* tasks. After fine-tuning, accuracy on *Missing Function* rises significantly to **39.00%** (Mistral-seed) and **35.00%** (GPT-OSS-seed).

4. **Superiority over SOTA Baselines:** Data from ToolWeave yields superior performance on the BFCL benchmark compared to the ToolFlow and ToolDial baselines. The poor performance of ToolDial (e.g., 3.75% on Llama-70B) is because its training data lacks the implicit multi-step dependencies and error recovery patterns required to succeed on a live, executable benchmark. ToolFlow's high-level plans, on the other hand, lack the fine-grained control needed by non-frontier LLMs like `gpt-oss-120b`, resulting in hallucinated or inferior dialogues that fail to train the model effectively. This is reflected in the results: when using the same synthesis LLM, Llama-3.1-70B achieves **33.25%** accuracy with ToolWeave data, compared to only **21.00%** with ToolFlow data. This demonstrates that our modular, fine-grained pipeline is critical for generating high-quality training data from non-frontier LLMs.

### 5.2.2 GENERALIZATION TO DIVERSE BENCHMARKS

To assess generalization, we evaluate models fine-tuned on ToolWeave and baseline data across three additional, diverse benchmarks. As shown in Table 5, ToolWeave-tuned models consistently outperform base models, while baseline-trained models sometimes underperform. On both ToolHop and CONFETTI, baseline models fail to extract necessary information from user queries and tool documentation, often overlooking critical details, asking redundant clarifying questions, or hallucinating arguments. In contrast, ToolWeave-trained models demonstrate marked improvements in these areas. These results highlight the robustness of ToolWeave-generated data, indicating that the learned capabilities generalize effectively across tasks rather than being limited to specific benchmarks.

## 6 CONCLUSION

We presented *ToolWeave*, a modular framework for creating fully synthetic multi-turn tool-calling datasets using non-frontier models. Our method builds everything from the ground up—from synthetic APIs to final multi-turn dialogues. Our fine-grained framework provides the precise control needed to guide non-frontier LLMs, enabling them to generate high-quality, complex data that they would otherwise fail to produce. ToolWeave yields datasets that significantly improve LLM performance on multi-turn tool-calling benchmarks, outperforming existing pipelines. These results show that high-quality, license-friendly data can be produced at scale without depending on proprietary models. As future work, we plan to extend ToolWeave with fully executable synthetic APIs and broaden dialogue generation to capture richer agentic patterns such as ReAct (Yao et al., 2023).

Table 5: Comparison of model performance on API Bank, CONFETTI, and ToolHop.

| Model | API Bank Level 1 | API Bank Level 2 | Confetti | ToolHop |
|---|---|---|---|---|
| Llama-3.1-8B-Instruct | 62.91 | 59.46 | 20.55 | 10.05 |
| + FT on ToolDial (GPT-4o-seed) | 59.15 | 52.70 | 23.12 | 9.55 |
| + FT on ToolFlow (GPT-OSS-seed) | 67.67 | 55.41 | 17.19 | 16.98 |
| **+ FT on ToolWeave (GPT-OSS-seed)** | **68.92** | **60.81** | **36.76** | **21.61** |
| Llama-3.1-70B-Instruct | 54.89 | 59.46 | 33.00 | 11.46 |
| + FT on ToolDial (GPT-4o-seed) | **71.93** | 40.54 | 11.46 | 5.83 |
| + FT on ToolFlow (GPT-OSS-seed) | 65.66 | 60.81 | 21.34 | 12.76 |
| **+ FT on ToolWeave (GPT-OSS-seed)** | 71.18 | **64.86** | **45.45** | **22.51** |
| Phi-4 | 34.09 | 25.68 | 13.44 | 10.55 |
| + FT on ToolDial (GPT-4o-seed) | 55.64 | 44.59 | 13.64 | 5.13 |
| + FT on ToolFlow (GPT-OSS-seed) | 44.11 | 39.19 | 16.80 | 6.03 |
| **+ FT on ToolWeave (GPT-OSS-seed)** | **71.68** | **62.16** | **33.60** | **12.56** |

## REPRODUCIBILITY STATEMENT

We have taken several concrete steps to ensure the reproducibility of our results. ToolWeave's core contribution lies in its ability to generate high-quality, fully synthetic multi-turn tool-calling data using open and non-frontier models. This makes the generated data license-friendly for both academic and commercial use. We believe this is a valuable contribution to the community, and have made efforts to clearly document our methodology and release all necessary resources. Section 3 outlines the overall pipeline, while Appendix B, C, D and E provides formal and detailed descriptions of each component. Appendix G includes hardware, software, and hyperparameter details used for model training, and Appendix H describes the evaluation benchmarks. Sample inputs and outputs for each component are shown in Appendix I. The prompts used throughout ToolWeave are provided in the supplementary materials. Finally, the source code and dataset will be released upon acceptance of the paper to facilitate verification and reuse.

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

# APPENDIX

## A    USE OF LARGE LANGUAGE MODELS

In preparing this paper, we made limited use of Large Language Models (LLMs) as writing assistance tools. Specifically, we used Grammarly[2] for spell checking and grammar correction to improve clarity and readability. In line with the conference policy, we confirm that LLMs were not used for generating research ideas or making any substantive contributions to the scientific content of this work.

## B    TOOL GRAPH SYNTHESIZER DETAILS

### B.1    PROMPTS USED

All prompts used in the tool graph synthesis stage are provided in a zip file included with the supplementary data. After extraction, they can be found in the `tool_graph_synthesizer` directory.

### B.2    DOMAINS FOR TOOL GRAPH SYNTHESIS

The 20 domains used to generate the synthetic API pool are:

| | | |
|---|---|---|
| Agriculture | Human Resources | Real Estate |
| Customer Support | Insurance | Retail |
| Cybersecurity | Legal Services | Supply Chain |
| E-commerce | Logistics | Telecommunications |
| Education | Manufacturing | Tourism |
| Energy | Marketing | Transportation |
| Film Industry | Online Banking | |

### B.3    SAMPLE TOOL GRAPH

Figure 2 illustrates a representative tool graph for the E-commerce domain. The nodes represent discrete tools (APIs), connected by directed edges that signify data dependencies; specifically, a parameter from the output schema of a predecessor node maps to an input parameter of the successor (e.g., `zone_policy` from `get_geo_rules` acts as an input for `set_mode`).

The graph topology highlights three distinct execution patterns, corresponding to the Tool-Graph traversal strategies discussed in Section 3.2:

- **Linear Chains:** Standard sequential dependencies connected by any type of edge, such as `get_order` → `set_mode` → `ship_local` → `save_track`.

- **Fan-in-Fan-out (Dashed):** Illustrated by the sequence `get_order` → {`check_stock`, `calc_risk`} → `sync_status`. Here, the initial tool provides inputs to a parallel stage of tools that are independent of one another, the outputs of which collectively feed into a common successor.

- **Conditional Branching (Green):** Represented by the `set_mode` node, where the downstream execution path (`ship_local` vs. `ship_intl`) is determined dynamically based on the output value of the decision node.

---

[2] https://app.grammarly.com/

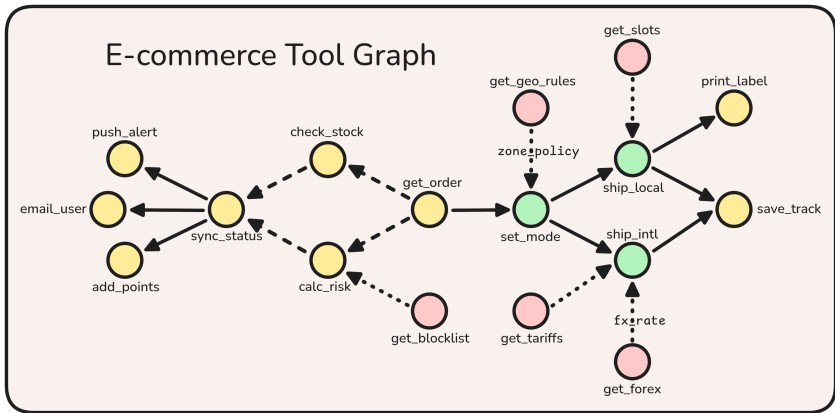

Figure 2: E-commerce tool graph demonstrating linear, fan-in-fan-out (dashed), and conditional (green) dependency patterns. Red nodes indicate auxiliary tools providing specific configuration parameters, such as foreign exchange rates or zone policies, to the primary workflow.

## B.4 DETAILED ALGORITHMS FOR TOOL GRAPH SYNTHESIZER

This section expands upon the Tool Graph Synthesizer process from Section 3.1. While the description in the main paper presents the high-level methodology, here we provide the main algorithm, the synthesis plan, and detailed sub-algorithms for key steps like domain context construction and API refinement.

---

**Algorithm 1:** Tool Graph Synthesis

---

**Input:** Domain descriptor $D$, synthesis plan $P$, LLM $L$, embedding model $E$
**Output:** API pool $\mathcal{A}$, Tool Graph $\mathcal{G}$
Initialize $\mathcal{A} \leftarrow \emptyset, \mathcal{G} \leftarrow \emptyset$;
$C \leftarrow \text{CONSTRUCTDOMAINCONTEXT}(D)$ ;                 // See Appendix B.6.1
**foreach** *step* $s \in P$ **do**
    Generate $n_s$ candidate APIs with $L$ using domain context $C$ and prompt from step $s$;
    Parse and validate candidates syntactically;
    Deduplicate candidates against $\mathcal{A}$ using embeddings from $E$ and structural checks;
    Refine and semantically validate schemas (defaults, enums, nested objects, etc.);
    Add accepted APIs to $\mathcal{A}$;
Construct Tool Graph $\mathcal{G}_{tool}$ by validating data-flows between APIs in $\mathcal{A}$;
**return** $\mathcal{A}, \mathcal{G}_{tool}$

---

## B.5 SYNTHESIS PLAN

Our synthesis plan acts as a curriculum for the API-generating LLM. It specifies a sequence of generation steps, each with a unique prompt template and a target number of APIs. This design is extensible and allows us to control the breadth and depth of the final API pool. Our implementation consists of five main stages:

1. **Seed Generation:** This initial stage generates a small batch of fundamental, "entry-point" APIs for the domain (e.g., search_product in e-commerce).
2. **Entity Expansion:** This stage uses the entities extracted from Wikidata (see Appendix B.6.1) to generate new APIs that specifically cover domain-relevant entities, ensuring breadth.
3. **Schema Enrichment:** This stage focuses on depth. It takes existing, simple APIs and iteratively increases their structural complexity by adding nested objects, enums, default values, and required flags to mimic real-world enterprise-grade schemas.

4. **Connection Discovery:** It generates new APIs that plausibly connect existing APIs, creating data-flow paths (e.g., a `get_product_details` API that takes a `product_id` from `search_product`).
5. **Pattern Expansion:** This final stage diversifies the API pool by generating parallel variations of existing APIs (e.g., different ways to search, like `search_by_name`, `search_by_category`, etc.).

For reproducibility, we include the synthesis plan used in our experiments below. Prompt paths have been abstracted for clarity.

```
{
    "steps": [
        {"name": "Seed Generation", "num_to_generate": 8},
        {"name": "Entity Expansion", "num_to_generate": 8},
        {"name": "Schema Enrichment", "num_to_generate": 5},
        {"name": "Connection Discovery", "num_to_generate": 8},
        {"name": "Pattern Expansion", "num_to_generate": 5}
    ]
}
```

### B.6 DETAILED SUB-ALGORITHMS

The main generation algorithm (Algorithm 1) relies on several key sub-routines, which are detailed below.

#### B.6.1 DOMAIN CONTEXT CONSTRUCTION

---
**Algorithm 2:** CreateDomainContext

---
**Input:** Domain name string $d$
**Output:** Domain context $C$
Resolve $d$ to canonical Wikidata entity $QID$;
Retrieve Wikipedia summary $T_{wiki}$ for $d$;
Extract structured facts $F_{wikidata}$ (classes, subclasses, properties) from $QID$;
Assemble $C \leftarrow (T_{wiki}, F_{wikidata})$;
**return** $C$

---

#### B.6.2 API REFINEMENT

---
**Algorithm 3:** RefineAPI

---
**Input:** Candidate API $a$, current API set $\mathcal{A}$
**Output:** Refined API $a'$ or $\emptyset$
Check if $a$ is duplicate of any in $\mathcal{A}$ (lexical, structural, semantic);
**if** *duplicate* **then**
 └ **return** $\emptyset$
Remove echo parameters (outputs attributes that are same as inputs);
Enrich schema: infer enums, default values, required fields;
Normalize parameter names and types;
Paraphrase descriptions for stylistic variance;
**return** *refined API* $a'$

---

### B.6.3 Connection Graph Construction

---

**Algorithm 4:** ConstructToolGraph

---

**Input:** API set $\mathcal{A}$
**Output:** Connection graph $\mathcal{G}$
Initialize graph $\mathcal{G} = (V = \mathcal{A}, E = \emptyset)$;
**foreach** *ordered pair* $(a_i, a_j) \in \mathcal{A} \times \mathcal{A}$ **do**
    Identify candidate parameter matches $(p_{out}, p_{in})$ ;    // By param name matches
    Validate match using LLM judgment;
    **if** *valid* **then**
        Add edge $(a_i.p_{out} \to a_j.p_{in})$ to $E$;

**return** $\mathcal{G}$

---

Matching parameters between APIs by exact name is justified in our case because, during API synthesis, the prompts instruct the model to design new APIs such that connections to existing APIs occur only through parameters with identical names.

## B.7 API Quality Metric Definitions

This section provides the formal definitions for the metrics used to evaluate the quality of the synthetic APIs in Section 4.1. In all definitions, $A = \{a_1, a_2, \ldots, a_n\}$ represents the set of $n$ APIs in a given domain.

### 1. Interconnectivity ($IC$)

The average number of input parameters per API that can be filled by an output from another API in the graph. This measures the data-flow potential.

$$IC = \frac{\sum_{i=1}^{n} |I_{a_i} \cap O|}{n}$$

Where:

- $O = \bigcup_{i=1}^{n} O_{a_i}$: The set of all unique output parameters from all APIs.
- $I_{a_i}$: The set of input parameters for API $a_i$.
- $|I_{a_i} \cap O|$: The count of input parameters for $a_i$ that match any output from the entire API set $A$.

### 2. Complex API Use ($CAU$)

The proportion of APIs that use at least one complex parameter type (i.e., "object" or "array"), measuring schema depth.

$$CAU = \frac{\sum_{i=1}^{n} C(a_i)}{n}$$

Where the indicator function $C(a_i)$ is defined as:

$$C(a_i) = \begin{cases} 1 & \text{if API } a_i \text{ has at least one parameter of type "object" or "array"} \\ 0 & \text{otherwise} \end{cases}$$

### 3. Required Parameter Ratio (RPR)

The average proportion of an API's input parameters that are marked as "required", measuring schema strictness.

$$RPR = \frac{1}{n} \sum_{i=1}^{n} \frac{|I_{\text{req}}(a_i)|}{|I_{\text{total}}(a_i)|}$$

Where $I_{\text{req}}(a_i)$ is the set of required input parameters for API $a_i$, and $I_{\text{total}}(a_i)$ is the set of all input parameters for $a_i$.

4. LONGEST CHAIN LENGTH

The length of the longest simple path (a path with no repeated nodes) in the *Tool Graph*. This metric quantifies the maximum number of sequential, multi-step operations possible in a single dialogue.

## C  GOAL GENERATION IMPLEMENTATION DETAILS

This section provides the low-level implementation details for the *Goal Generation* stage (Section 3.2), including the algorithms and heuristics used to extract, score, and synthesize goals.

### C.1  PROMPTS USED

All prompts used in the tool graph sampler and goal generation stage are provided in a zip file included with the supplementary data. After extraction, they can be found in the `tool_graph_sampler` directory.

### C.2  PATTERN EXTRACTION ALGORITHMS

The main paper outlines three pattern classes. Here is how they are discovered:

1. **Linear Paths:** These are found using a bounded *beam search* (Norvig & Russel, 2002). The search explores paths from a start node, expanding a "beam" of the top-$k$ candidates at each depth.
2. **Fan-in / Fan-out:** These patterns are found by analyzing node connectivity. For a given `start_node`, we find its successors. We then find the "common children" of combinations of these successors using set intersection. This allows us to identify points where parallel data flows (fan-out) later merge (fan-in).
3. **Conditional Branches:** These are identified by scanning an API's output schema. We scan for output fields that can act as a logical predicate, specifically those typed as `boolean`, `enum`, or other simple types, while filtering out non-conditional fields like IDs.

### C.3  PATH SCORING AND RANKING

To rank the paths found by our algorithms, we use a combination of a hybrid score and a diversification algorithm.

**Hybrid Heuristic Score:**  The `final_score` for any given path is a weighted sum of three components, and is calculated as:

$$S_{\text{final}} = w_l \cdot (S_{\text{coherence}} + S_{\text{relevance}}) + w_d \cdot S_{\text{dataflow}} + w_b \cdot S_{\text{length}}$$

Where:

- $S_{\text{coherence}}$ and $S_{\text{relevance}}$ are *LLM-based ratings* (from $-2$ to $+2$) where the LLM judges the quality of a synthesized goal for the provided tool path.
- $S_{\text{dataflow}}$ is the *semantic dataflow score*, calculated by checking cosine similarity between embeddings of output and input parameters of sequential tools.
- $S_{\text{length}}$ is a *length bonus* to reward longer, more complex paths.
- $w_l$, $w_d$, and $w_b$ are the respective weights for LLM rating, dataflow, and length, set as hyperparameters (we use $w_l = 0.5$, $w_d = 0.8$, $w_b = 0.3$ in our experiments).

**Ranking and Embedding Model:** To ensure a diverse set of goals, we rank the top-scoring linear paths using *Maximal Marginal Relevance (MMR)* (Carbonell & Stewart, 1999). This balances the `final_score` (relevance) with the cosine dissimilarity from already-selected goals (diversity). In our experiments, we use a value of $0.7$ for $\lambda$. All embeddings for MMR and semantic dataflow are computed using the `all-MiniLM-L6-v2` model from the *SentenceTransformers* (Reimers & Gurevych, 2019) library.

## C.4 GOAL GENERATION ALGORITHM

---

**Algorithm 5:** Domain-level Goal Generation

---

**Input:** Tool graph for domain $\mathcal{G}_{tool}$, beam width $B$, max depth $D_{\max}$, selection size $K$, MMR weight $\lambda$, LLM $L$, embedding model $E$

**Output:** Goal set $\mathcal{G}_{goals}$ (JSON objects: {`tool_path`, `pattern_type`, `goal_text`, `metadata`})

Initialize $\mathcal{G}_{goals} \leftarrow \emptyset, \mathcal{S} \leftarrow \emptyset,$ where $\mathcal{S}$ is the set of selected tool paths;

$S_{linear} \leftarrow \text{BEAMSEARCH}(\mathcal{G}_{tool}, B, D_{\max}, L)$;

$S_{linear} \leftarrow \text{TOPKMMR}(S_{linear}, K, \lambda, E)$;

$S_{fan} \leftarrow \text{FINDFANINFANOUT}(\mathcal{G}_{tool})$;

$S_{cond} \leftarrow \text{FINDCONDITIONAL}(\mathcal{G}_{tool})$;

$S \leftarrow S_{linear} \cup S_{fan} \cup S_{cond}$;

**foreach** $p \in S$ **do**

$\quad prompt \leftarrow \text{BUILDGOALPROMPT}(p, \text{PATTERNTYPE}(p))$;

$\quad goal\_text \leftarrow L.\text{GENERATE}(prompt)$;

$\quad obj \leftarrow \text{BUILDJSON}\big(p, \text{PATTERNTYPE}(p), goal\_text, \text{METADATA}(p)\big)$;

$\quad \mathcal{G}_{goals} \leftarrow \mathcal{G}_{goals} \cup \{obj\}$;

**return** $\mathcal{G}_{goals}$

---

## C.5 EXAMPLE GOAL SYNTHESIS PROMPT

The final step, 'Goal synthesis via LLM', uses different prompts for each pattern type. The prompt provides the LLM with the tool sequence and their schemas, and asks it to generate a natural-language goal.

# D DIALOGUE PLANNER IMPLEMENTATION DETAILS

This section provides the low-level implementation details for the *Dialogue Planner* stage (Section 3.3), including the algorithm and the logic used to partition tool paths, generate subgoals, plan parameters, and inject interaction patterns.

## D.1 PROMPTS USED

The prompts relevant to this planner stage are provided in a supplementary zip file. After extracting the zip file, they can be found in the `plan_generator` directory.

## D.2 DIALOGUE PLANNER ALGORITHM

Algorithm 6 summarizes the high-level process, highlighting subgoal generation, speaker sequencing, parameter planning, and clarification insertion.

---

**Algorithm 6:** Dialogue Planning

---

**Input:** Tool path $P$, pattern type $T \in \{\texttt{linear}, \texttt{fan}, \texttt{conditional}\}$, overall goal text $G$, relevant tool schemas $\Sigma$, goal metadata $M$, clarification probability $p_{clar}$, LLM for partitioning and subgoal synthesis $L$

**Output:** Ordered plan $\mathcal{D}$ (step-level JSON objects with keys: $\texttt{step\_idx}$, $\texttt{role}$, $\texttt{subgoal}$, $\texttt{tools}$, $\texttt{params}$, $\texttt{metadata}$)

Initialize partition list $\mathcal{P}^* \leftarrow \text{PARTITIONTOOLPATH}(P, T, \Sigma, L)$, subgoal text list $\mathcal{S} \leftarrow [\,]$ ;

**foreach** *partition* $p \in \mathcal{P}^*$ **do**
    $\mathcal{S}.\text{APPEND}(\text{SYNTHESIZESUBGOAL}(p, G, L))$ ;

$\mathcal{D} \leftarrow [\,]$ ;

**for** $i \leftarrow 1$ **to** $|\mathcal{P}^*|$ **do**
    $p \leftarrow \mathcal{P}^*[i]$ , $s \leftarrow \mathcal{S}[i]$ ;
    $\text{Params} \leftarrow \text{EXTRACTTOOLPARAMS}(p, \Sigma)$ ;
    $\text{Params} \leftarrow \text{Params} \setminus \text{UPSTREAMTOOLOUTPUTS}(\mathcal{P}^*, i - 1, \Sigma)$ ;
    $(\text{ReqParams}, \text{ClarParams}) \leftarrow \text{SPLITPARAMS}(\text{Params}, p_{clar})$ ;
    $\mathcal{D}.\text{APPEND}(\text{CREATEUSERTURN}(i, s, p, \text{ReqParams}))$ ;
    **if** $\text{ClarParams} \neq \emptyset$ **then**
        $\mathcal{D}.\text{APPEND}(\text{CREATEASSISTANTCLARIFICATIONTURN}(i, \text{ClarParams}))$ ;
        $\mathcal{D}.\text{APPEND}(\text{CREATEUSERCLARIFICATIONTURN}(i, \text{ClarParams}))$ ;
    **foreach** *tool* $t \in p$ **do**
        $\text{ToolParams} \leftarrow \text{EXTRACTTOOLPARAMS}([t], \Sigma)$ ;
        $\mathcal{D}.\text{APPEND}(\text{CREATEASSISTANTTOOLCALLTURN}(i, t, \text{ToolParams}, M))$ ; // Goal
         metadata $M$ attaches decision variables for conditionals
    $\mathcal{D}.\text{APPEND}(\text{CREATEASSISTANTSUMMARYTURN}(i, p))$ ;

**return** $\mathcal{D}$

---

### D.3 TOOL PATH PARTITIONING

The PARTITIONTOOLPATH function uses different strategies based on the pattern type:

- **Linear:** We use an LLM-based approach. The prompt provides the full tool path and their schemas, and asks the LLM to split them into coherent segments, based on logical groupings and dataflow, while also ensuring meaningful multi-step tool sequences.
- **Fan-in / Fan-out:** We use a deterministic graph-based approach. We identify the branching point (fan-out) and the merging point (fan-in) in the path, and create partitions accordingly. The intermediate tools between these points are grouped into either segment, and all possible valid partitions are generated to allow for diverse dialogue structures.
- **Conditional:** Based on the value of the decision variable (from goal metadata $M$), the second tool is chosen from the available branches. The first tool and this chosen branch form a single partition to ensure the conditional logic is preserved in the dialogue, while also generating high-quality multi-step dialogues.

### D.4 SUBGOAL SYNTHESIS

The SYNTHESIZESUBGOAL function uses an LLM prompt that provides the tool schemas in the partition and the overall goal text $G$. The LLM is instructed to generate a concise, user-friendly subgoal that accurately reflects the purpose of the tools in the partition, while ensuring it is distinct from other subgoals in the dialogue.

### D.5   Parameter Planning and Clarification Insertion

The parameter planning and clarification insertion process involves several steps to ensure that the dialogue captures realistic user-assistant interactions.

#### D.5.1   Parameter Preparation

The EXTRACTTOOLPARAMS function retrieves all input parameters for the tools in the current partition. The UPSTREAMTOOLOUTPUTS function identifies any parameters that can be automatically filled from outputs of tools in previous partitions, and these are removed from the required parameters.

#### D.5.2   Clarification Insertion

The SPLITPARAMS function randomly selects a subset of the remaining parameters to be clarified, based on the clarification probability $p_{clar}$. This introduces variability and simulates real-world scenarios where users may not provide all necessary information upfront.

#### D.5.3   Speaker Sequencing and Turn Creation

The turn creation functions add well-formatted JSON objects to the dialogue plan, specifying the role and the associated content (subgoal, tool calls, parameter requests, clarifications, summaries). Following is the structure of these turns:

- `CreateUserTurn`: User turn with subgoal and parameters to provide upfront.
- `CreateAssistantClarificationTurn`: Assistant turn asking for clarification on missing parameters required for the tool calls in the current partition.
- `CreateUserClarificationTurn`: User turn providing values for the requested parameters.
- `CreateAssistantToolCallTurn`: Assistant turn invoking a tool with the provided parameters and source hints for each parameter, either from the user or from upstream tool outputs.
- `CreateAssistantSummaryTurn`: Assistant turn summarizing the results of the tools in the current partition.

## E   DIALOGUE SYNTHESIZER Implementation Details

This section provides the low-level implementation details for the *Dialogue Synthesizer* stage (Section 3.4), including the overall algorithm and the logic used to build context for each role across agents.

### E.1   Prompts Used

The prompts associated with each agent role in this stage, including the post-processing state for user message paraphrasing, as well as those relevant to the dialogue synthesis and refinement stages, are provided in the `dialogue_synthesizer` directory, available after extracting the supplementary zip file.

### E.2   Dialogue Synthesis Algorithm

Algorithm 7 summarizes the high-level process, highlighting context construction, dynamic role-based agent selection, response generation, and memory updates.

---

**Algorithm 7:** Plan-Driven Dialogue Synthesis

---

**Input:** Ordered plan $\mathcal{D}$, tool schemas $\Sigma$, language models for agents $L$
**Output:** Dialogue transcript $\mathcal{T}$
Initialize memory $\mathcal{M} \leftarrow \emptyset$, transcript $\mathcal{T} \leftarrow [\,]$;
**foreach** *step* $d \in \mathcal{D}$ **do**
   $ctx \leftarrow$ BUILDCONTEXT$(\mathcal{M}, d, \Sigma)$ ;      `// Get memory and plan step`
   $role \leftarrow d.\text{role}$;
   $agent\_prompt \leftarrow$ SELECTPROMPT$(role)$ ;   `// Prompt with agent's role`
   $response \leftarrow L.\text{GENERATE}(agent\_prompt, ctx)$ ;    `// Utterance/tool call`
   $\mathcal{T}.\text{APPEND}(response)$;
   $\mathcal{M} \leftarrow$ UPDATEMEMORY$(\mathcal{M}, response, d)$ ;  `// Update memory with output`
**return** $\mathcal{T}$

---

# F DIALOGUE POST-PROCESSING

## F.1 DIALOGUE ERROR INJECTION

To rigorously evaluate and improve the robustness of models trained on ToolWeave data, we implemented a systematic error injection pipeline. This pipeline deterministically introduces schema violations, logical ordering errors, and tool confusion scenarios to force the model to learn error recovery in simulated real-world settings. Algorithm 8 details the high-level injection logic, while Algorithm 9 illustrates the complex logic required to synthesize cascading failure scenarios.

---

**Algorithm 8:** Dialogue Error Injection

---

**Input:** Dialogue set $\mathcal{D}$, Tool Schemas $\mathcal{S}$, Injection Probability $p_{inject}$, Similarity Matcher $M$
**Output:** Augmented Dialogue set $\mathcal{D}_{aug}$
Initialize $\mathcal{D}_{aug} \leftarrow \emptyset$;
**foreach** *dialogue* $d \in \mathcal{D}$ **do**
   Add original $d$ to $\mathcal{D}_{aug}$;
   **if** $Random(0, 1) < p_{inject}$ **then**
      $d_{error} \leftarrow$ DeepCopy$(d)$;
      $injected \leftarrow$ False;
      `// Attempt Complex Logical Errors (30% chance)`
      **if** $Random(0, 1) < 0.3$ **then**
         **if** INJECTCASCADINGFAILURE*($d_{error}$, $\mathcal{S}$) is Success* **then**
            $injected \leftarrow$ True;
         **else if** INJECTOUTOFORDERERROR*($d_{error}$, $\mathcal{S}$) is Success* **then**
            $injected \leftarrow$ True;
         **else if** INJECTWRONGTOOLERROR*($d_{error}$, $\mathcal{S}$, $M$) is Success* **then**
            $injected \leftarrow$ True;
      `// Attempt Simple Parameter Errors (70% chance or if`
      `   complex failed)`
      **if** *not injected* **then**
         INJECTSCHEMAERROR$(d_{error}, \mathcal{S})$;
      **if** *error injected successfully* **then**
         Mark $d_{error}$ as modified;
         Add $d_{error}$ to $\mathcal{D}_{aug}$;
**return** $\mathcal{D}_{aug}$

---

---

**Algorithm 9:** Inject Cascading Failure

---

**Input:** Dialogue $d$, Tool Schemas $\mathcal{S}$
**Output:** Modified Dialogue $d'$, Success boolean
Identify long multi-step sequences $T_{seq}$ (len $\geq 3$) in $d$;
**if** $T_{seq}$ *is empty* **then**
$\quad \llcorner$ **return** $d$, *False*
Select random sequence $S \in T_{seq}$;
`// 1.  Simulate Reverse-Order Execution`
Initialize failure_turns $\leftarrow \emptyset$;
**for** $i \leftarrow length(S)$-1 ***down to** 1* **do**
$\quad \mid \quad step_{curr} \leftarrow S[i]$, $step_{prev} \leftarrow S[i-1]$;
$\quad \mid \quad$ Identify param $p_{dep}$ in $step_{curr}$ dependent on $step_{prev}$;
$\quad \mid \quad$ Create turn $t_{call}$: Call $step_{curr}$ **missing** $p_{dep}$;
$\quad \mid \quad$ Create turn $t_{err}$: Error "Prerequisite step not completed/Missing input";
$\quad \llcorner \quad$ Append $(t_{call}, t_{err})$ to failure_turns;
`// 2.  Inject Full Recovery`
Construct new history: $history_{original} + failure_{turns} + S_{original}$;
Update $d'.conversations$ and re-index $d'.plan$ with error steps;
**return** $d'$, *True*

---

**Additional Error Injection Modules:** Beyond the cascading failures detailed in Algorithm 9, our pipeline includes:

- **Out-of-Order Execution:** We simulate scenarios where the model attempts to execute a tool call before its dependencies are met. Specifically, we identify dependent tool pairs and inject a premature call to the downstream tool (missing the dependent parameter) prior to the prerequisite tool. The model receives a missing dependency error and must recover by executing the sequence in the correct order.
- **Semantic Confusion (Wrong Tool):** We utilize a hybrid similarity matcher (Sentence-BERT (Reimers & Gurevych, 2019) + Levenshtein distance (Lcvenshtcin, 1966)) to identify semantically similar but incorrect tools (e.g., `search_ticket` vs. `get_ticket_details`). We inject a turn where the model mistakenly calls the confusable tool, receives an unhelpful response, and must self-correct.
- **Schema Violations:** We deterministically mutate valid tool calls to violate JSON schema constraints. This includes removing required parameters (`MISSING_PARAM`), violating type constraints (e.g., passing an integer for a string field), or using invalid enum values. The model is trained to read the resulting standardized error message and retry with the correct schema.
- **Missing Function Recovery:** We simulate scenarios where a necessary tool is initially absent from the provided context. The algorithm identifies a scheduled tool call in the plan and temporarily "hides" it. We inject an Assistant turn refusing the request due to the missing tool, followed by a User turn that explicitly provides the missing tool's schema JSON. The model is trained to recognize the new tool definition in the context window and immediately proceed with the correct tool call.

## G  FINE-TUNING HYPERPARAMETERS

We fine-tune all models on 8 NVIDIA A100 SXM4 80GB GPUs using `QLoRA` via the `LLaMA-Factory` framework[3]. The primary hyperparameters used for each model are detailed in Table 6. Common settings across all models include a `learning_rate` of $5.0 \times 10^{-5}$, `bf16` mixed-precision, 4-bit `bnb` quantization, and a `cosine` learning rate scheduler.

## H  EVALUATION BENCHMARK DETAILS

This section provides additional details on the benchmarks used in our experiments (Section 5).

---

[3] `https://github.com/hiyouga/LLaMA-Factory`

Table 6: Key fine-tuning hyperparameters for all models.

| Hyperparameter | Llama-3.1-8B-Instruct | Llama-3.1-70B-Instruct | Phi-4 |
|---|---|---|---|
| *Model Details* | | | |
| quantization_bit | 4 | 4 | 4 |
| flash_attn | fa2 | fa2 | fa2 |
| *LoRA Configuration* | | | |
| lora_rank | 32 | 8 | 16 |
| lora_alpha | 64 | 32 | 32 |
| lora_dropout | 0.05 | 0.05 | 0.05 |
| lora_target | all | all | all |
| *Training Details* | | | |
| num_train_epochs | 1.0 | 1.0 | 1.0 |
| warmup_ratio | 0.05 | 0.1 | 0.1 |
| per_device_train_batch_size | 1 | 1 | 1 |
| gradient_accumulation_steps | 8 | 2 | 8 |
| gradient_checkpointing | false | true | false |

**BFCL-V3:** The BFCL-V3 (Patil et al., 2025a) benchmark evaluates multi-turn tool-calling across four challenging categories: *Base*, *Missing Function*, *Missing Parameter*, and *Long Context*. It focuses on conversational scenarios that require clarification, refusal, and robustness. Performance is reported as execution accuracy in a live, executable environment, with credit awarded only when all steps in a dialogue are correctly resolved.

**API Bank:** The API Bank (Li et al., 2023) benchmark consists of 8 domains, 73 tools, and 314 dialogues. We evaluate on Level 1 ("Call") and Level 2 ("Retrieve+Call"). As tool retrieval is not the focus of this paper, we provide the set of required tools to the model for both levels. The evaluation is conducted in an off-policy, teacher-forced manner, where the model must predict the next step given the correct history.

**CONFETTI:** The CONFETTI (Alkhouli et al., 2025) benchmark is a multi-turn, multi-step conversational benchmark with 109 human-simulated conversations covering 86 tools. A key characteristic is its explicit inclusion of conversational complexities, such as follow-ups, goal correction, goal switching, ambiguous goals, and over- or under-filled parameters. We focus on evaluating the turn-level tool-calling accuracy.

**ToolHop:** The ToolHop (Ye et al., 2025) benchmark is specifically designed to evaluate multi-step tool use within a single turn. It comprises 995 queries and 3912 locally executable tools. Answering a query requires the implicit decomposition of the request and sequential invocation of multiple tools where the output of one tool is the input for another. ToolHop evaluates models in a live environment, testing both planning and error recovery, with accuracy computed based on the final answer.

# I TOOLWEAVE SAMPLE DATA - GOAL, PLAN, TOOLS, AND DIALOGUE

We present a detailed example from the customer support domain that illustrates a complex, multi-step dialogue. The example showcases the five available tools, the high-level goal constructed using these tools, the 15-step fine-grained plan, and the final multi-turn dialogue that realizes the plan. The following subsections present each component in turn.

## I.1 TOOLS LIST

**Tool 1: create_support_ticket**

```
{
  "name": "create_support_ticket",
  "description": "Create support ticket.",
  "parameters": {
    "type": "object",
    "properties": {
      "issue_description": {"type": "string", "description": "A
      ↪   detailed description of the issue."},
      "category": {"type": "string", "enum": ["technical",
      ↪   "billing", "general"], "default": "general"},
      "requester_id": {"type": "string", "description": "The
      ↪   unique identifier for the customer requesting
      ↪   support."},
      "urgency_level": {"type": "string", "enum": ["low",
      ↪   "medium", "high"], "default": "medium"}
    },
    "required": ["issue_description", "requester_id"]
  },
  "results": {
    "type": "object",
    "properties": {
      "ticket_id": {"type": "string", "description": "The unique
      ↪   identifier for the newly created ticket."},
      "creation_date": {"type": "string", "format":
      ↪   "date-time"},
      "status": {"type": "string", "enum": ["open",
      ↪   "in_progress", "resolved", "closed", "on_hold"]}
    }
  }
}
```

**Tool 2: get_ticket_details**

```
{
  "name": "get_ticket_details",
  "description": "Retrieve comprehensive details about a
  ↪   specific support ticket.",
  "parameters": {
    "type": "object",
    "properties": {
      "support_ticket_identifier": {"type": "string",
      ↪   "description": "The unique reference number of the
      ↪   ticket"}
    },
    "required": ["support_ticket_identifier"]
  },
  "results": {
    "type": "object",
    "properties": {
      "customer_id": {"type": "string"},
      "issue_description": {"type": "string"},
      "priority": {"type": "string", "enum":
      ↪   ["low","medium","high"]},
      "category": {"type": "string", "enum":
      ↪   ["technical","billing","general"]},
      "status": {"type": "string", "enum":
      ↪   ["open","in_progress","resolved","closed","on_hold"]},
      "creation_date": {"type": "string", "format":
      ↪   "date-time"},
      "last_updated": {"type": "string", "format": "date-time"}
    }
  }
}
```

**Tool 3: search_tickets**

```json
{
  "name": "search_tickets",
  "description": "Find support tickets by criteria.",
  "parameters": {
    "type": "object",
    "properties": {
      "start_date": {"type": "string", "format": "date"},
      "end_date": {"type": "string", "format": "date"},
      "user_account_id": {"type": "string"},
      "issue_type": {"type": "string", "enum":
      ↪  ["technical","billing","general"], "default":
      ↪  "general"},
      "ticket_state": {"type": "string", "enum":
      ↪  ["open","in_progress","resolved","closed"],
      ↪  "default": "open"},
      "urgency_level": {"type": "string", "enum":
      ↪  ["low","medium","high"], "default": "medium"}
    },
    "required": ["user_account_id"]
  },
  "results": {
    "type": "object",
    "properties": {
      "tickets": {
        "type": "array",
        "items": {
          "type": "object",
          "properties": {
            "ticket_id": {"type": "string"},
            "issue_description": {"type": "string"},
            "creation_date": {"type": "string", "format":
            ↪  "date-time"},
            "last_updated": {"type": "string", "format":
            ↪  "date-time"}
          }
        }
      }
    }
  }
}
```

**Tool 4: escalate_ticket_to_specialist**

```
{
  "name": "escalate_ticket_to_specialist",
  "description": "Elevates a support ticket to a specialist
  ↪  team.",
  "parameters": {
    "type": "object",
    "properties": {
      "specialist_team": {"type": "string", "enum":
      ↪  ["technical","billing","legal","management"]},
      "urgency_level": {"type": "string", "enum":
      ↪  ["low","medium","high","critical"], "default":
      ↪  "medium"},
      "attachments": {"type": "array", "items": {"type":
      ↪  "object"}},
      "support_case_id": {"type": "string"},
      "specialist_notes": {"type": "string"}
    },
    "required":
    ↪  ["specialist_team","support_case_id","specialist_notes"]
  },
  "results": {
    "type": "object",
    "properties": {
      "escalation_id": {"type": "string"},
      "escalation_date": {"type": "string", "format":
      ↪  "date-time"},
      "status": {"type": "string", "enum":
      ↪  ["pending","in_progress","resolved","rejected"]}
    }
  }
}
```

**Tool 5: update_escalation_status**

```
{
  "name": "update_escalation_status",
  "description": "Modify ticket escalation status.",
  "parameters": {
    "type": "object",
    "properties": {
      "status": {"type": "string", "enum":
      ↪  ["pending","in_progress","resolved","rejected"]},
      "ticket_escalation_id": {"type": "string"}
    },
    "required": ["status","ticket_escalation_id"]
  },
  "results": {
    "type": "object",
    "properties": {
      "last_updated": {"type": "string", "format": "date-time"}
    }
  }
}
```

## I.2 HIGH LEVEL GOAL

> **Goal**
>
> *Create a support ticket, review its details, and escalate it to a specialist team for urgent resolution.*

## I.3 PLAN

> **Multi-step Plan**
>
> **Step 1 (USER_UTTERANCE)**
> Utterance: *I need to create a support ticket and get all the details about it.*
> Provided params:
>
> ```
> {
>     "create_support_ticket.requester_id":
>     "$user_provided_$create_support_ticket.requester_id"
> }
> ```
>
> **Step 2 (ASSISTANT_CLARIFICATION)**
> Parameter names:
>
> ```
> ["create_support_ticket.issue_description"]
> ```
>
> **Step 3 (USER_RESPONSE_TO_CLARIFICATION)**
> Provides params:
>
> ```
> {
>     "create_support_ticket.issue_description":
>     "$user_provided_$create_support_ticket.issue_description"
> }
> ```
>
> **Step 4 (CALL_TOOL: create_support_ticket)**
> Parameters:
>
> ```
> {
>     "create_support_ticket.issue_description":
>     "$user_provided_$create_support_ticket.issue_description",
>     "create_support_ticket.requester_id":
>     "$user_provided_$create_support_ticket.requester_id"
> }
> ```
>
> **Step 5 (CALL_TOOL: get_ticket_details)**
> Parameters:
>
> ```
> {
>     "get_ticket_details.support_ticket_identifier":
>     "$create_support_ticket.ticket_id"
> }
> ```

**Step 6 (ASSISTANT_RESPONSE_TOOL)**
Summarizes tools:

```
["create_support_ticket", "get_ticket_details"]
```

Outputs:

```
[
    "create_support_ticket.ticket_id",
    "create_support_ticket.creation_date",
    "create_support_ticket.status",
    "get_ticket_details.customer_id",
    "get_ticket_details.issue_description",
    "get_ticket_details.priority",
    "get_ticket_details.category",
    "get_ticket_details.status",
    "get_ticket_details.creation_date",
    "get_ticket_details.last_updated"
]
```

**Step 7 (USER_UTTERANCE)**
Utterance: *Can you search for the ticket I just created?*

**Step 8 (CALL_TOOL: search_tickets)**
Parameters:

```
{
  "search_tickets.user_account_id":
  ↪  "$get_ticket_details.customer_id",
  "search_tickets.issue_type": "$get_ticket_details.category",
  "search_tickets.ticket_state": "$create_support_ticket.status"
}
```

**Step 9 (ASSISTANT_RESPONSE_TOOL)**
Summarizes tools:

```
["search_tickets"]
```

Outputs:

```
[
    "search_tickets.tickets[].ticket_id",
    "search_tickets.tickets[].issue_description",
    "search_tickets.tickets[].creation_date",
    "search_tickets.tickets[].last_updated"
]
```

**Step 10 (USER_UTTERANCE)**
Utterance: *Now, please escalate this ticket to a specialist team for urgent resolution and update the escalation status.*
Provided params:

```
{
    "escalate_ticket_to_specialist.specialist_team":
    "$user_provided_$escalate_ticket_to_specialist.specialist_tea↵
    ↪   m"
}
```

### Step 11 (ASSISTANT_CLARIFICATION)
Parameter names:

```
[
    "escalate_ticket_to_specialist.specialist_notes",
    "update_escalation_status.status"
]
```

### Step 12 (USER_RESPONSE_TO_CLARIFICATION)
Provides params:

```
{
    "escalate_ticket_to_specialist.specialist_notes":
    "$user_provided_$escalate_ticket_to_specialist.specialist_n↵
    ↪   otes",
    "update_escalation_status.status":
    "$user_provided_$update_escalation_status.status"
}
```

### Step 13 (CALL_TOOL: escalate_ticket_to_specialist)
Parameters:

```
{
    "escalate_ticket_to_specialist.specialist_team":
    "$user_provided_$escalate_ticket_to_specialist.specialist_t↵
    ↪   eam",
    "escalate_ticket_to_specialist.support_case_id":
    "$create_support_ticket.ticket_id",
    "escalate_ticket_to_specialist.specialist_notes":
    "$user_provided_$escalate_ticket_to_specialist.specialist_n↵
    ↪   otes"
}
```

### Step 14 (CALL_TOOL: update_escalation_status)
Parameters:

```
{
    "update_escalation_status.status":
    "$user_provided_$update_escalation_status.status",
    "update_escalation_status.ticket_escalation_id":
    "$escalate_ticket_to_specialist.escalation_id"
}
```

### Step 15 (ASSISTANT_RESPONSE_TOOL)
Summarizes tools:

```
["escalate_ticket_to_specialist", "update_escalation_status"]
```

Outputs:

```
[
    "escalate_ticket_to_specialist.escalation_id",
    "escalate_ticket_to_specialist.escalation_date",
    "escalate_ticket_to_specialist.status",
    "update_escalation_status.last_updated"
]
```

## I.4  DIALOGUE

**Multi-turn Dialogue**

**System:** Current time: 2025-08-27 21:24:05.

**User:** I need to create a support ticket for an issue I'm having with my account login. I keep getting an error message saying 'Invalid credentials' even though I'm sure I'm entering the correct username and password. I've tried resetting my password multiple times, but the issue persists. I need this resolved as soon as possible because I can't access any of my account features.

**Assistant:** Could you please provide your requester ID so I can create a support ticket for your account login issue?

**User:** My requester ID is cust123456789.

**Assistant:**
```
create_support_ticket({"issue_description": "I keep
getting an error message saying 'Invalid credentials'
even though I'm sure I'm entering the correct username
and password. I've tried resetting my password
multiple times, but the issue persists. I need this
resolved as soon as possible because I can't access
any of my account features.", "requester_id":
"cust123456789", "urgency_level": "high", "category":
"general"})
```
**Tool:**
```
{"ticket_id": "tkt987654321", "creation_date":
"2025-08-27T21:24:05Z", "status": "open"}
```
**Assistant:**
```
get_ticket_details({"support_ticket_identifier":
"tkt987654321"})
```
**Tool:**
```
{"customer_id": "cust123456789", "issue_description":
"I keep getting an error message saying 'Invalid
credentials' even though I'm sure I'm entering the
correct username and password. I've tried resetting
my password multiple times, but the issue persists. I
need this resolved as soon as possible because I can't
access any of my account features.", "priority":
"high", "category": "general", "status": "open",
"creation_date": "2025-08-27T21:24:05Z",
"last_updated": "2025-08-27T21:24:05Z"}
```

**Assistant:** I've created a support ticket for your account login issue. Your ticket ID is tkt987654321, created on August 27, 2025, and it is currently open. The ticket details confirm that the issue is related to receiving an 'Invalid credentials' error message despite entering the correct username and password, and that you've attempted to reset your password multiple times without success. The ticket is categorized as 'general' with high priority. The ticket status remains open, and it was last updated on August 27, 2025. Our support team will address this as soon as possible.

**User:** Can you search for the ticket I just created?

**Assistant:**
```
search_tickets({"user_account_id":  "cust123456789",
"issue_type":  "general", "ticket_state":  "open",
"urgency_level":  "medium"})
```

**Tool:**
```
{"tickets":  [{"ticket_id":  "tkt987654321",
"issue_description":  "I keep getting an error message
saying 'Invalid credentials' even though I'm sure I'm
entering the correct username and password.  I've
tried resetting my password multiple times, but the
issue persists.  I need this resolved as soon as
possible because I can't access any of my account
features.", "creation_date":  "2025-08-27T21:24:05Z",
"last_updated":  "2025-08-27T21:24:05Z"}]}
```

**Assistant:** I found your open support ticket with ID tkt987654321. It was created on August 27, 2025, regarding the 'Invalid credentials' error you're experiencing despite entering the correct username and password. The issue description mentions that you've tried resetting your password multiple times without success. This ticket was last updated on August 27, 2025, and our support team is working to resolve it as quickly as possible.

**User:** Please escalate my ticket to a specialist team for urgent resolution. Here are some additional details: I've been experiencing this issue for the past three days, and it's preventing me from accessing critical account features. I've tried all the standard troubleshooting steps, including clearing my browser cache and trying different devices, but nothing has worked. I suspect there might be an issue with the account authentication system itself.

**Assistant:** To which specialist team would you like me to escalate your ticket for urgent resolution?

**User:** Please escalate my ticket to the technical team and set the escalation status to in_progress.

**Assistant:**
```
escalate_ticket_to_specialist({"specialist_team":
"technical", "support_case_id":  "tkt987654321",
"specialist_notes":  "I've been experiencing this issue
for the past three days, and it's preventing me from
accessing critical account features.  I've tried all
the standard troubleshooting steps, including clearing
my browser cache and trying different devices, but
nothing has worked.  I suspect there might be an issue
with the account authentication system itself.",
"attachments":  [], "urgency_level":  "high"})
```

**Tool:**
```
{"escalation_id":  "esc987654321", "escalation_date":
"2025-08-27T21:24:05Z", "status":  "in_progress"}
```

**Assistant:**
```
update_escalation_status({"status":  "in_progress",
"ticket_escalation_id":  "esc987654321"})
```

**Tool:**
`{"last_updated": "2025-08-27T21:24:05Z"}`

**Assistant:** Your ticket has been successfully escalated to the technical team. The escalation ID is esc987654321, created on August 27, 2025, with a status of "in_progress". This escalation was last updated on August 27, 2025. The technical team will now prioritize resolving your account login issue, considering the additional details you've provided about the problem's duration and your troubleshooting attempts.

## J    EXAMPLE DIALOGUE GENERATED VIA TOOLFLOW

This example dialogue highlights a common issue in dialogues generated via the ToolFlow pipeline, where the tool agent hallucinates a tool response without the assistant agent providing an actual tool call, resulting in a fabricated tool reply (as seen in the first assistant response and the first tool response).

---

**Multi-turn Dialogue**

**User:** I've just bought a new field and have the map outline. Could you define its boundary with this GeoJSON: `{"type":"Feature","geometry":{"type":"Polygon",` `"coordinates":[[[30.0,10.0],[40.0,40.0],[20.0,40.0],` `[10.0,20.0],[30.0,10.0]]]},"properties":{}}`? The field ID is "field-12345".

**Assistant:** Sure, defining the boundary for field field-12345 now.

**Tool:**
`{"field_boundary_id": "boundary-67890", "status":` `"defined"}`

**Assistant:** Great! The boundary for field field-12345 has been successfully defined (status: defined). I can now pull a short-term weather forecast for the next week for this field. Would you like me to retrieve that for you?

**User:** Yes, please pull the short-term weather forecast for the next week for field-12345.

**Assistant:** Sure! To get the short-term weather forecast I'll need the farm identifier that corresponds to this field. Could you provide the farm ID (or let me know if the field ID can be used in place of a farm ID)?

**User:** Sure, the farm ID is farm-abc123.

**Assistant:**
`get_weather_forecast({"farm_identifier":` `"farm-abc123"})`

**Tool:**
`{"temperature_celsius": 22.3, "precipitation_mm":` `0.0, "wind_speed_kph": 14.5, "condition": "sunny"}`

**Assistant:** Here's the forecast I've retrieved for tomorrow at your field field-12345 (farm-abc123):

---

# K  SOME EXAMPLE DIALOGUES FROM TOOLDIAL TRAINING SET

The following examples illustrate some of the issues why LLMs fine-tuned on ToolDial data exhibit inferior performance on the BFCL-V3 benchmark (see Table 4).

Example K.1 shows that models trained on such data struggle with parameter elicitation and error recovery—the assistant neither asks for the missing hotel IP address nor attempts to resolve the misunderstanding after the user's correction.

## K.1  EXAMPLE 1

---

**Multi-turn Dialogue**

**User:** Can you help me find detailed information about local businesses, such as restaurants and cafes, near my hotel? I only have the hotel's IP address and need to find its geo-location first to get the coordinates.

**Assistant:** I can help you find detailed information about local businesses near your hotel. Would you like me to proceed with finding nearby businesses?

**User:** No, that's not what I need.

**Assistant:** I'm sorry, but I cannot fulfill your request due to insufficient information. Please provide more details or try a different query.

---

## K.2  EXAMPLE 2

Example K.2 illustrates two issues. The user's initial request is somewhat unrealistic and unlikely to occur in a real-world setting. The assistant's clarification questions are phrased using internal parameter names rather than natural, intuitive language that a user would expect.

---

**Multi-turn Dialogue**

**User:** What is the population within a defined area around a city, given only the city's name and its country?

**Assistant:** Would it be appropriate to retrieve the population data within a defined area around the city you mentioned?

**User:** That would be great. Sure, that sounds good.

**Assistant:** Please tell me the radius of the circular area around the coordinates and the coordinates of the geographic point to search nearby.

**User:** I can provide the radius, which is 300, but I don't know the coordinates.

**Assistant:** Please provide the namePrefixDefaultLangResults for the city.

**User:** Sure! The namePrefixDefaultLangResults for the city is San Francisco.

**Assistant:**
```
[{ "name":  "cities", "arguments":  {
"namePrefixDefaultLangResults":  "San Francisco" } } ]
```

---

**Tool:**
```
{ "error":  "", "response":  { "data":  [{ "id":
5392171, "wikiDataId":  "Q123456", "type":  "CITY",
"city":  "San Francisco", "name":  "San Francisco",
"country":  "United States", "countryCode":  "US",
"region":  "California", "regionCode":  "CA",
"regionWdId":  "Q99", "population":  883305,
"coordinates":  "[37.7749, -122.4194]"}, { "id":
5902958, "wikiDataId":  "Q123457", "type":  "CITY",
"city":  "San Jose", "name":  "San Jose", "country":
"United States", "countryCode":  "US", "region":
"California", "regionCode":  "CA", "regionWdId":
"Q99", "population":  1035317, "coordinates":
"[37.3382, -121.8863]"}, { "id":  5322760,
"wikiDataId":  "Q123458", "type":  "CITY", "city":
"San Diego", "name":  "San Diego", "country":  "United
States", "countryCode":  "US", "region":
"California", "regionCode":  "CA", "regionWdId":
"Q99", "population":  1423851, "coordinates":
"[32.7157, -117.1611]"}], "links":  { "next":
"/v1/geo/cities?offset=60&limit=50" }, "metadata":  {
"currentOffset":  10, "totalCount":  150 } } }
```

**Assistant:** Please choose one of the following cities: San Francisco, San Jose, San Diego.

**User:** I choose San Francisco.

**Assistant:**
```
get_population({ "radius":  "300", "coordinate":
"[37.7749, -122.4194]" })
```

**Tool:**
```
{ "error":  "", "response":  { "radius":  300,
"population":  12000, "description":  "The estimated
resident population within a 300-meter radius of the
specified coordinates in San Francisco is
approximately 12,000 individuals.  This figure is
based on the latest demographic data and reflects the
density of the area, which includes residential
buildings, parks, and commercial establishments.",
"coordinates":  [37.7749, -122.4194] } }
```

**Assistant:** The estimated resident population within a 300-meter radius of the specified coordinates in San Francisco is approximately 12,000 individuals. This figure is based on the latest demographic data and reflects the density of the area, which includes residential buildings, parks, and commercial establishments.

**User:** Thank you for the information. Goodbye!

**Assistant:** You're welcome! Goodbye!

