# OpenReview forum: "TOOLWEAVE: FINE-GRAINED AND CONTROLLABLE SYNTHETIC DATA GENERATION FOR MULTI-TURN TOOL CALLING WITH NON-FRONTIER LLMS"
_ICLR.cc/2026/Conference — Submitted to ICLR 2026_

### Official Review · Reviewer_JXqN · 2025-10-22

**Soundness:** 3
**Presentation:** 3
**Contribution:** 3
**Rating:** 4
**Confidence:** 4

**Summary:**

This paper presents ToolWeave, a fine-grained and controllable data synthesis pipeline for multi-turn tool calling. Through detailed design and task decomposition, the authors demonstrate that non-frontier LLMs (e.g., gpt-oss-120b) can generate high-quality training data. The API candidates are fully synthesized from a single domain name, thereby alleviating potential licensing issues. Furthermore, different graph traversal methods applied to the generated tool graph, in combination with subgoal decomposition, enable more fine-grained generation.

Data analysis indicates that the synthesized data is of high quality, and experimental results show that models trained on this data achieve improved performance across multiple multi-turn benchmarks, validating the effectiveness of the proposed pipeline.

**Strengths:**

1. Good motivation for proposing pipelines that are suitable for non-frontier LLMs.
2. Subgraph sampling with different methods ensures the diversity and compatibility for tool candidates.
3. Detailed analysis on the data statistics validates the data quality.

**Weaknesses:**

1. While I appreciate the efforts made to enable non-frontier LLMs to generate multi-turn tool-calling data samples, I am not fully convinced that the proposed pipeline effectively advances this goal. For example, synthesizing a group of tools capable of supporting complex queries (i.e., multiple combinations of dependent tool usages) based solely on a single domain name would be challenging even for frontier LLMs. Moreover, both subgoal generation and agent role assignment depend heavily on the LLMs’ overall planning ability.

2. The paper lacks an ablation study to reveal the contributions of each proposed component. This issue is closely related to the first weakness. I do not see any analysis or experiment that demonstrates how the proposed components reduce the reliance on LLM capability, thereby enabling non-frontier LLMs to generate high-quality data.

3. Results on more challenging and widely adopted benchmarks, such as $\tau$-bench and $\tau^2$-bench, are expected.

4. While the authors claim that being “fully synthesized” is an advantage, I disagree. More recent work (e.g., MCP-related benchmarks) emphasizes the importance of developing LLM capabilities in real environments. Relying on LLMs to simulate tools inevitably introduces errors or inconsistencies in the generated tool outputs, resulting in data that is at high risk of deviating from real-world scenarios.

**Questions:**

1. How do you get the initial domain name for tool synthesis? If merely hand-crafted, how to scale up for generation?
2. It seems that the quality of the generated data highly depends on the JSON plan generated by the planner. What is your concerns determining that putting so many details (even the role of each step is maintained) in the JSON plan?
3. Have you tried your pipeline with frontier LLMs like GPT-5 or with more smaller LLMs like Qwen3-32B?

---

> ### Author Response · Authors · 2025-11-23
> **Response to Reviewer JXqN (1/5)**
>
> We thank the reviewer for validating our motivation to empower non-frontier LLMs and for recognizing the diversity ensured by our subgraph sampling strategy and the comprehensive analysis of data quality.
>
> ---
>
> **Weakness 1:** The pipeline's effectiveness is questioned because complex steps (tool synthesis, planning) rely heavily on advanced LLM capabilities, which challenges its use for non-frontier models.
>
> **Response:** We thank the reviewer for this critical assessment. We fully agree that asking a non-frontier model to perform "one-shot" complex planning or holistic API synthesis would result in failure. However, **ToolWeave is explicitly designed to bypass this limitation by decomposing these complex tasks into atomic, verifiable operations** that are well within the capabilities of open-weights models. We address the specific concerns below:
>
> **1. Feasibility of Tool Synthesis (Decomposition vs. One-Shot):** The reviewer correctly notes that synthesizing a coherent, interconnected toolset from a domain name is a difficult task. We address this by avoiding single-shot generation entirely. As detailed in Section 3.1, we employ an "Iterative, Curriculum-Driven Synthesis" approach. Instead of asking the LLM to "generate all APIs of a domain in One-Shot" we decompose the task into progressively simpler stages: *Seed Generation $\\rightarrow$ Entity Expansion $\\rightarrow$ Schema Enrichment $\\rightarrow$ Connection Discovery*. By feeding the verified output of stage $N$ as context to stage $N+1$, the LLM performs only incremental additions (e.g., "create an API that links to this specific output") to the set of APIs. This reduces the complexity to a level manageable for non-frontier models.
>
> **2. Clarification on Planning and Role Assignment:** We wish to clarify a misunderstanding regarding the Fine-Grained Plan Generator (Section 3.3). The reviewer expresses concern that role assignment depends on the LLM's planning ability. We address this by clarifying the pipeline's order of operations, which rigorously decouples logical planning from text generation:
>
> - **Simplified Subgoal Generation:** First, we reduce the planning horizon via **partitioning**. This splits a long tool sequence into atomic segments (e.g., `[[A], [B, C]]`) using graph traversal rules rather than LLM intuition. The LLM's role is restricted to generating a text description for each isolated segment: a simple summarization task that does not require complex reasoning.
> - **Deterministic Role Assignment:** Once partitioned, the assignment of agent roles is not decided by an LLM. The planner algorithmically expands each subgoal into a fixed sequence of turns (*User Request $\\rightarrow$ Optional Clarification $\\rightarrow$ Tool Execution $\\rightarrow$ Summary*) based on the partition's metadata. This guarantees a valid speaker trajectory, ensuring the LLM never has to "guess" the correct role or transition.
>
> By reducing the workflow to a series of simple tasks, *incremental addition, local summarization, and algorithmic sequencing*, we enable non-frontier models to construct complex, multi-turn dialogues without requiring the reasoning capabilities of frontier models.

---

> > ### Author Response · Authors · 2025-11-23
> > **Response to Reviewer JXqN (2/5)**
> >
> > **Weakness 2:** No ablation study is provided to show how components reduce reliance on LLM capability.
> >
> > **Response:** We thank the reviewer for this constructive suggestion. To empirically demonstrate how our pipeline stages **reduce reliance on raw LLM capability** and justify their necessity, we conducted a comprehensive ablation study using Llama-3.1-8B-Instruct on the BFCL-V3 benchmark.
> >
> > **1. Necessity of Dialogue Post-Processing:** Table 1 isolates the impact of the post-processing stage. The results show that injecting robustness patterns (e.g., error recovery, missing functions) yields a substantial **+6.26% absolute gain** in Multi-Turn Accuracy. This confirms that the post-processor is essential for bridging the gap between "idealized" synthetic data and realistic execution scenarios, a capability frontier models might simulate intrinsically but open models require assistance to master.
> >
> > **Table 1: Performance impact of the post-processing (postproc) stage.**
> > Evaluated on Llama-3.1-8B-Instruct, post-processing produces large and consistent gains across all metrics.
> >
> > \\[
> > \\begin{array}{|c|c|c|c|c|c|}
> > \\hline
> > \\textbf{Setting} & \\textbf{Multi-Turn Acc} & \\textbf{Base} & \\textbf{MissFunc} & \\textbf{MissParam} & \\textbf{LongCtx} \\\\
> > \\hline
> > \\text{ToolWeave (GPT-OSS seed) without postproc} & 13.62 & 14.50 & 14.50 & 13.50 & 12.00 \\\\
> > \\hline
> > \\text{ToolWeave (GPT-OSS seed) with postproc} & \\mathbf{19.88} & \\mathbf{23.00} & \\mathbf{21.50} & \\mathbf{15.50} & \\mathbf{19.50} \\\\
> > \\hline
> > \\text{Absolute performance gain} & +6.26 & +8.50 & +7.00 & +2.00 & +7.50 \\\\
> > \\hline
> > \\end{array}
> > \\]
> >
> > **2. Necessity of Structured Sampling and Fine-Grained Planning:** Table 2 validates the architecture of the generation pipeline itself (with post-processing disabled for fair comparison). Evaluated on Llama-3.1-8B-Instruct.
> >
> > - **Impact of Structured Sampler:** Replacing our motif-based sampler with a random walk (Row 2 In Table 2 below) degrades accuracy from 13.62% to 12.25%, demonstrating that coherent, structured goals are superior to random tool sequences.
> > - **Impact of Fine-Grained Planner (Critical for Non-Frontier Models):** Row 3 shows the result of using our structured goals but generating the dialogue via the baseline method (ToolFlow), which lacks our fine-grained JSON planning. Performance collapses to **7.50%**, effectively the same as the ToolFlow baseline. **This confirms that the Fine-Grained Planner is the critical enabling component that reduces reliance on model capability**. Without this step-by-step scaffolding, the non-frontier model lacks the raw reasoning power to convert even high-quality goals into valid data.
> >
> > **Table 2: Performance comparison across settings.**
> >
> > *Ablations via controlled component replacement. All settings exclude post-processing for comparability. We evaluate (1) replacing the structured sampler with random sampling while keeping ToolWeave's planner and synthesizer, and (2) replacing ToolWeave's planner+synthesizer with ToolFlow given the same structured goals.*
> >
> > \\[
> > \\begin{array}{|c|c|c|c|c|c|}
> > \\hline
> > \\textbf{Setting} & \\textbf{Multi-Turn Acc} & \\textbf{Base} & \\textbf{MissFunc} & \\textbf{MissParam} & \\textbf{LongCtx} \\\\
> > \\hline
> > \\text{ToolWeave (no postproc)} & \\textbf{13.62} & \\textbf{14.50} & \\textbf{14.50} & \\textbf{13.50} & \\textbf{12.00} \\\\
> > \\hline
> > \\text{Random sampling + ToolWeave (Plan + synth)} & 12.25 & 13.00 & 14.00 & \\textbf{13.50} & 8.50 \\\\
> > \\hline
> > \\text{Structured sampling + goals } \\rightarrow \\text{ ToolFlow} & 7.50 & 9.50 & 7.50 & 5.00 & 8.00 \\\\
> > \\hline
> > \\text{ToolFlow baseline (GPT-OSS seed)} & 7.62 & 11.00 & 4.50 & 5.50 & 9.50 \\\\
> > \\hline
> > \\end{array}
> > \\]
> >
> > ---
> > \
> > **Weakness 3:** Evaluation is needed on more challenging and widely adopted benchmarks (e.g., $\\tau$-bench and $\\tau^2$-bench).
> >
> > **Response:** We thank the reviewer for suggesting these challenging benchmarks. We recognize the value of $\\tau$-bench and $\\tau^2$-bench for evaluating agentic capabilities. We note that these benchmarks rely on frontier-class models (e.g., GPT-5) to serve as high-fidelity user simulators. While our work primarily focuses on enabling open-source workflows independent of such proprietary dependencies, we agree that testing on these widely adopted standards would further strengthen our evaluation. We are currently arranging the necessary resources to access these frontier simulators and conduct the evaluation. Should these experiments be completed within the rebuttal period, we will report the results in a future revision.

---

> > > ### Author Response · Authors · 2025-11-23
> > > **Response to Reviewer JXqN (3/5)**
> > >
> > > **Weakness 4:** Relying on fully synthesized tools risks deviations from real-world execution, challenging the claimed advantage.
> > >
> > > **Response:** We value the reviewer's perspective on the importance of real-world environments (such as MCP benchmarks) and the potential risks of simulation-based training. We agree that minimizing the gap between simulation and reality is paramount. However, we respectfully argue that our "fully synthesized" approach does not sacrifice realism; rather, it allows us to systematically construct a training environment that is often richer and more robust than available real-world datasets. We achieve this through two mechanisms:
> > >
> > > **1. Structural Complexity Matching (Section 4.1):** Contrary to the concern that synthetic tools are simplistic or inconsistent, our empirical analysis shows they match or exceed the complexity of real-world baselines. In **Section 4.1**, we compare our generated toolset against 4,474 real-world APIs from RapidAPI. Our synthetic APIs demonstrate:
> > > - **Higher Parameter Density:** An average of **3.6** parameters per API (vs. 2.4 for real APIs).
> > > - **Greater Depth:** A **Complex API Use (CAU)** score of **23.6%**, compared to just 1.0% for real-world APIs, indicating significantly higher usage of nested objects and arrays.
> > >
> > > This demonstrates that our synthesis pipeline creates schemas that are not merely "consistent," but rigorously challenging, preventing the model from overfitting to simplistic "toy" tools.
> > >
> > > **2. Simulating Real-World Non-Determinism (Section 3.5):** The reviewer rightly points out that simulation can lack the friction of real execution (e.g., errors or side effects). We explicitly mitigate this risk via **Dialogue Post-Processing (Section 3.5)**. Rather than relying on a perfect "happy path" simulation, we deterministically inject noise to mimic real-world instability:
> > >
> > > * **Error Recovery:** We introduce erroneous tool calls with realistic failure messages to train the model in correction strategies.
> > > * **Missing-Function Scenarios:** We simulate environments where necessary tools are temporarily unavailable, training the model to refuse requests or find workarounds.
> > > * **Out-of-Order Execution:** We model non-linear workflow variations.
> > >
> > > By explicitly engineering these "imperfections" into the dataset, we ensure the model is trained to handle the messiness of real-world execution without being constrained by the limitations of accessing live commercial APIs. **We have updated the paper to include a new Appendix F.1 (referenced in Sec 3.5)**, which details the formal algorithms for **Dialogue Error Injection** component.
> > >
> > > ---
> > > \
> > > **Question 1:** How do you get the initial domain name for tool synthesis? If merely hand-crafted, how to scale up for generation?
> > >
> > > **Response:** For the experiments in this paper, the 20 domains listed in Appendix B.2 were manually selected to ensure a diverse cross-section of industries for robust evaluation.
> > >
> > > However, this manual selection is not a limitation for scaling. As described in Section 3.1, the only input required to bootstrap our entire pipeline is a simple "domain string" (e.g., `customer_support`). This makes scaling trivial and programmatic: one can easily automate domain generation by querying Wikidata or Wikipedia.
> > > Furthermore, this design simplifies targeted generation: if a user requires data for a specific, niche domain, they simply need to provide the corresponding Wikipedia page title. The pipeline will then automatically retrieve the relevant narrative summaries and entity structures to synthesize a full, domain-specific dataset.

---

> ### Author Response · Authors · 2025-11-23
> **Response to Reviewer JXqN (4/5)**
>
> **Question 2:** Concerns exist regarding the reliance on the detailed JSON plan and embedding excessive step-level information.
>
> **Response:** We thank the reviewer for this insightful question. ToolWeave's JSON plan intentionally introduces **low-level, fine-grained details**, and this design choice is driven by the limitations of non-frontier LLMs. High-level or loosely specified plans, as used in prior pipelines, routinely lead to drift, missing parameters, and broken tool chains when executed by weaker non-frontier LLMs. This granularity is intentional: it offloads most of the planning and reasoning burden from the non-frontier LLM, turning a difficult multi-step decision-making problem into a simpler slot-filling and natural language generation task.
>
> The main concern behind this design is the risk of over-rigidity: highly detailed plans could in principle produce overly scripted dialogues. We mitigate this in two ways:
> **1. Decoupling logic from expression:** The plan encodes only the interaction logic (roles, data flow, tool calls), while the Multi-Agent Dialogue Synthesizer produces free-form natural language, preserving fluency and variability.
> **2. Post-processing for diversity:** Paraphrasing and utterance rewrites (Sec. 3.5) inject lexical variation on top of the consistent logical structure.
>
> To validate the design choice of high-granularity planning, we compare the full ToolWeave pipeline against an ablation where the **Fine-Grained Plan Generator** is replaced by the baseline ToolFlow planner (while keeping the high-quality structured goals). The drastic drop in performance (from 13.62% to 7.50%) confirms that the fine-grained deterministic plan is the critical scaffolding that enables the LLMs to execute these goals.
>
> **Necessity of Fine-Grained Planning**
>
> *All settings exclude post-processing for comparability. Evaluated on Llama-3.1-8B-Instruct.*
> \\[
> \\begin{array}{|c|c|c|c|c|c|}
> \\hline
> \\textbf{Setting} & \\textbf{Multi-Turn Acc} & \\textbf{Base} & \\textbf{MissFunc} & \\textbf{MissParam} & \\textbf{LongCtx} \\\\
> \\hline
> \\text{ToolWeave (no postproc)} & 13.62 & 14.50 & 14.50 & 13.50 & 12.00 \\\\
> \\hline
> \\text{Structured sampling + goals } \\to \\text{ ToolFlow's Monolithic planner} & 7.50 & 9.50 & 7.50 & 5.00 & 8.00 \\\\
> \\hline
> \\end{array}
> \\]

---

> > ### Author Response · Authors · 2025-11-23
> > **Response to Reviewer JXqN (5/5)**
> >
> > **Question 3:** Have you tried your pipeline with frontier LLMs like GPT-5 or with more smaller LLMs like Qwen3-32B?
> >
> > **Response:** We appreciate the reviewer's suggestion to evaluate the pipeline across different model scales. We have extended our experiments to test Mistral-Small-3.1-24B-Instruct (~24B parameters) as a synthesis LLMs, alongside our existing results for gpt-oss-120b and mistral-medium-2505.
> >
> > The results demonstrate that ToolWeave remains highly effective even when driven by a smaller 24B model. As shown in the table below, the Llama-3.1-8B model fine-tuned on data from the 24B seed achieves **20.00%** accuracy on BFCL-V3, which is competitive with the **19.88%** achieved using the 120B seed. This confirms that our pipeline's scaffolding allows even smaller, accessible models to generate high-quality training data.
> >
> > \\[
> > \\begin{array}{|l|c|c|c|c|c|}
> > \\hline
> > \\textbf{Setting} & \\textbf{Multi-Turn Acc} & \\textbf{Base} & \\textbf{MissFunc} & \\textbf{MissParam} & \\textbf{LongCtx} \\\\
> > \\hline
> > \\mathrm{Llama\\text{-}8B\\ +\\ FT\\ (ToolWeave\\ GPT\\text{-}OSS\\text{-}120B\\ seed)} & 19.88 & 23.00 & \\mathbf{21.50} & 15.50 & 19.50 \\\\
> > \\hline
> > \\mathrm{Llama\\text{-}8B\\ +\\ FT\\ (ToolWeave\\ Mistral\\text{-}Small\\text{-}24B\\ seed)} & 20.00 & 30.00 & 14.00 & 15.50 & 20.50 \\\\
> > \\hline
> > \\mathrm{Llama\\text{-}8B\\ +\\ FT\\ (ToolWeave\\ Mistral\\text{-}Medium\\ seed)} & \\mathbf{25.00} & \\mathbf{31.50} & 20.50 & \\mathbf{24.50} & \\mathbf{23.50} \\\\
> > \\hline
> > \\end{array}
> > \\]
> >
> > \\[
> > \\begin{array}{|l|c|c|c|c|c|}
> > \\hline
> > \\textbf{Setting} & \\textbf{Multi-Turn Acc} & \\textbf{Base} & \\textbf{MissFunc} & \\textbf{MissParam} & \\textbf{LongCtx} \\\\
> > \\hline
> > \\mathrm{Llama\\text{-}70B\\ +\\ FT\\ (ToolWeave\\ GPT\\text{-}OSS\\text{-}120B\\ seed)} & \\mathbf{33.25} & \\mathbf{37.50} & 35.00 & \\mathbf{32.00} & \\mathbf{28.50} \\\\
> > \\hline
> > \\mathrm{Llama\\text{-}70B\\ +\\ FT\\ (ToolWeave\\ Mistral\\text{-}Small\\text{-}24B\\ seed)} & 28.00 & 29.00 & 32.50 & 26.50 & 24.00 \\\\
> > \\hline
> > \\mathrm{Llama\\text{-}70B\\ +\\ FT\\ (ToolWeave\\ Mistral\\text{-}Medium\\ seed)} & 28.88 & 27.50 & \\mathbf{39.00} & 30.00 & 19.00 \\\\
> > \\hline
> > \\end{array}
> > \\]
> >
> > **Regarding the use of GPT-5:** We effectively performed this comparison in the paper (Table 4) by evaluating mistral-medium-2505 (a proprietary, frontier-class commercial model) alongside the open gpt-oss-120b. The results were illuminating: Llama-3.1-70B fine-tuned on data from the open model achieved 33.25% accuracy, actually outperforming the version trained on the commercial model seed (**28.88%**).
> >
> > This empirical evidence clarifies two key points:
> >
> > **1. Design is Essential:** The fact that the open model matches (or beats) the commercial model proves that **ToolWeave's fine-grained scaffolding is the primary driver of data quality**, effectively bridging the capability gap between open and frontier models.
> >
> > **2. Diminishing Returns of Frontier Models:** ToolWeave is engineered specifically to mitigate the weaknesses of non-frontier models by breaking synthesis into small, verifiable steps. As a result, stronger LLMs provide diminishing returns within our architecture. Because ToolWeave already supplies the reasoning structure that frontier models typically provide, we do not expect GPT-5 to meaningfully change the conclusions. While using GPT-5 may slightly improve fluency or lexical variation, we do not expect a qualitative jump in data quality, because the pipeline already controls the reasoning and structure. Furthermore, generating our full dataset using commercial frontier models is estimated to cost $\\$$400 (GPT-5-mini) to $\\$$1000 (GPT-5) per iteration, creating a prohibitive barrier for reproducible open research compared to using open-weights models.
> >
> > **That said, we are trying to arrange resources to extend our evaluation to GPT-5. Should these experiments be completed within the rebuttal period, we will report them in a future revision.**

---

> > > ### Comment · Reviewer_JXqN · 2025-11-26
> > >
> > > Thank you for the response! I appreciate the efforts for the additional experiment results. I have raised the score accordingly.

---

> ### Author Response · Authors · 2025-11-27
> **Response to Reviewer JXqN's comment**
>
> We sincerely thank the reviewer for the constructive feedback and for raising the score. We appreciate your engagement and are glad the additional results helped clarify our contributions.
>
> To answer your question regarding the usage of frontier models as data synthesis LLMs, we integrated **GPT-5-Mini** into the ToolWeave pipeline to empirically test the necessity of our design when using frontier models. *Note:* All the prompts and hyperparameters are fixed throughout the synthesis LLM ablations to ensure a fair comparison.
>
> The results in the table below show a modest improvement (**+2.00% overall**), validating our aforementioned hypothesis that stronger LLMs provide **diminishing returns** within our architecture that is built around the capabilities of non-frontier LLMs. Because ToolWeave already supplies the deterministic reasoning structure (planning, state tracking), GPT-5-mini primarily contributes fluency rather than a qualitative jump in logic.
>
> \\[
> \\begin{array}{|l|c|c|c|c|c|}
> \\hline
> \\textbf{Setting} & \\textbf{Multi-Turn Acc} & \\textbf{Base} & \\textbf{MissFunc} & \\textbf{MissParam} & \\textbf{LongCtx} \\\\
> \\hline
> \\text{ToolWeave (GPT-OSS-seed)} & 19.88 & 23.00 & \\mathbf{21.50} & 15.50 & 19.50 \\\\
> \\hline
> \\text{ToolWeave (GPT-5-mini-seed)} & \\mathbf{21.88} & 23.00 & 21.00 & \\mathbf{24.00} & 19.50 \\\\
> \\hline
> \\end{array}
> \\]
>
> **Complexity of GPT-5-mini generated data:** GPT-5-mini synthesized significantly richer schemas (Avg. **5.01 parameters/tool** vs 3.60). Because our training examples include the full definitions of all available tools in the context, these richer schemas resulted in much **longer contexts averaging ~20.9k tokens/sample** (vs ~11.6k).
>
> We maintained a standardized 16k training context window across all experiments to ensure a controlled comparison within a fixed compute budget. Due to this constraint, **60% of the GPT-5-mini generated examples were dropped entirely** (vs. negligible loss for GPT-OSS). This results in a performance lower bound because the model was trained on only a fraction of the available high-quality data. We are currently optimizing context length (e.g., pruning unused tools) which we expect will further improve performance. We are also extending these experiments to **Llama-3.1-70B-Instruct**.
>
> **Benefits of ToolWeave:** This experiment reinforces the advantages of our non-frontier approach:
> * **Prohibitive Cost:** Generating this iteration of dataset generation with GPT-5-mini cost us nearly **$400**. Scaling this to the bigger GPT-5 model would increase costs by ~5x, creating a barrier for reproducible research.
> * **Commercial Viability:** Most of the existing frameworks rely on frontier models like GPT-4/5 to synthesize data, which often prohibits commercial deployment. ToolWeave enables the creation of tool calling data using **open-source non-frontier models** using entirely **license-friendly** workflows.
> * **Data Sovereignty:** Our approach allows users to **synthesize sensitive domains locally**, resolving privacy constraints associated with exposing proprietary schemas to third-party providers.
>
> We hope these clarifications address the reviewer's concerns, and we appreciate the opportunity to improve the paper.

---

> > ### Comment · Reviewer_JXqN · 2025-11-28
> >
> > Thanks for the additional results! I have no further questions.

---

> ### Author Response · Authors · 2025-11-30
> **Updated results for frontier model ablation (GPT-5-mini)**
>
> We sincerely appreciate your time and thoughtful review of our work. Thank you for the valuable feedback, which has helped us improve the paper. Also we sincerely thank the reviewer for raising the score.
>
> ---
> As promised, we have completed the context optimization experiments for **GPT-5-mini**. We also include our existing **Mistral-Medium (Commercial)** baseline from the paper and a new **Mistral-Small-24B (Open)** ablation we did during rebuttal to provide a comprehensive comparison across model scales and licensing types.
>
> **1. Experimental Setup:**
> In our original training setup, we passed the schemas of all tools in the domain to the model's context, which caused 60% of samples to exceed the 16k token limit and be dropped. To resolve this, we modified the training setup to include only the tools used in the conversation and eight distractor tools from the same domain. This ensured that 100% of the GPT-5-mini generated examples fit within the 16k token window.
>
> **2. Updated Empirical Results:**
>
> We fine-tuned Llama-3.1-8B and 70B on data generated by four different synthesis models.
>
> \\[
> \\begin{array}{|c|c|c|c|c|c|}
> \\hline
> \\textbf{8B Model Results} & \\textbf{Acc} & \\textbf{Base} & \\textbf{MissFunc} & \\textbf{MissParam} & \\textbf{LongCtx} \\\\
> \\hline
> \\text{ToolWeave (Mistral-Small-24B) [Open]} & 20.00 & 30.00 & 14.00 & 15.50 & 20.50 \\\\
> \\hline
> \\text{ToolWeave (GPT-OSS-120B) [Open]} & 19.88 & 23.00 & 21.50 & 15.50 & 19.50 \\\\
> \\hline
> \\text{ToolWeave (Mistral-Medium) [Commercial]} & \\mathbf{25.00} & \\mathbf{31.50} & 20.50 & \\mathbf{24.50} & \\mathbf{23.50} \\\\
> \\hline
> \\text{ToolWeave (GPT-5-mini) [Frontier]} & 23.12 & 25.00 & \\mathbf{22.50} & 24.00 & 21.00 \\\\
> \\hline
> \end{array}
> \\]
>
> \\[
> \\begin{array}{|c|c|c|c|c|c|}
> \\hline
> \\textbf{70B Model Results} & \\textbf{Acc} & \\textbf{Base} & \\textbf{MissFunc} & \\textbf{MissParam} & \\textbf{LongCtx} \\\\
> \\hline
> \\text{ToolWeave (Mistral-Small-24B) [Open]} & 28.00 & 29.00 & 32.50 & 26.50 & 24.00 \\\\
> \\hline
> \\text{ToolWeave (GPT-OSS-120B) [Open]} & 33.25 & 37.50 & 35.00 & \\mathbf{32.00} & 28.50 \\\\
> \\hline
> \\text{ToolWeave (Mistral-Medium) [Commercial]} & 28.88 & 27.50 & \\mathbf{39.00} & 30.00 & 19.00 \\\\
> \\hline
> \\text{ToolWeave (GPT-5-mini) [Frontier]} & \\mathbf{34.62} & \\mathbf{38.00} & 37.50 & 31.50 & \\mathbf{31.50} \\\\
> \\hline
> \\end{array}
> \\]
>
> **3. Key Findings from these experiments:**
> * **Architecture Compensates for Scale:** On the larger 70B model, our Open Source seed (gpt-oss-120b) outperforms the Commercial seed (mistral-medium) by +4.37% and trails the best Frontier model (gpt-5-mini) by only 1.37%. Notably, the small open-weight Mistral-Small-24B (28.00%) performs on par with the commercial Mistral-Medium (28.88%). This proves that ToolWeave's fine-grained scaffolding is the primary driver of data quality.
>
> * **Cost-Benefit Analysis:** Achieving this marginal gain required using a commercial API that costs significantly more per iteration compared to open-weights inference. This underscores the value of ToolWeave in enabling the community to generate SOTA-level, license-friendly data using accessible non-frontier models.

---

### Official Review · Reviewer_Kjdb · 2025-10-29

**Soundness:** 2
**Presentation:** 2
**Contribution:** 2
**Rating:** 4
**Confidence:** 4

**Summary:**

This work focuses on multi-turn tool-calling synthetic data generation pipelines, specifically targeting non-frontier LLMs. To this end, the authors propose ToolWeave, a controllable and modular framework that can synthesize high-quality multi-turn tool-calling datasets using license-friendly, non-proprietary models. ToolWeave consists of four key components: 1. synthesizing a  library of tools 2. sampling subgraphs from structural motifs 3. generating dialogue plans that decompose complex reasoning into small, executable steps, and 4. executing these plans, followed by post-processing to enhance diversity and robustness.

Experimental results show that LLMs fine-tuned on ToolWeave-generated data (using gpt-oss-120b) significantly outperform SOTA baselines (ToolFlow and ToolDial) on multi-turn tool-calling benchmarks such as BFCL-V3.

**Strengths:**

- The paper introduces the an open and modular framework for generating multi-turn, multi-tool dialogues
- The use of Wikipedia and Wikidata for domain grounding is both practical and elegant
- The system is methodically designed and evaluated across different tool-calling benchmarks, showing consistent improvements over competitive baselines.

**Weaknesses:**

- Terminology clarity and exposition
While most parts of the paper are well-written, some key terminologies lack concise explanations when first introduced.
For example, in the abstract (line 023), the authors mention “synthesize a Tool Graph of APIs,” but it is unclear what constitutes the nodes and edges of this graph. A brief definition before each new term would improve readability and accessibility.

- Motivation justification
The paper’s core motivation—to focus on non-frontier LLMs—needs stronger justification.
The mainstream approach for improving function-calling capabilities is to fine-tune on high-quality SFT data, and commercial models (e.g., GPT-5, BFCL multi-turn achieves 57.63) can already perform this task well.
Thus, if frontier models can readily produce such data, why is it necessary to design a specialized workflow for weaker models?
The motivation behind this work may be a false need.

- Evaluation ceiling and benchmark interpretation
The reported results on BFCL-V3 may not fully substantiate the claimed significance.
For instance, the best ToolWeave-tuned model (Llama-3.1-70B) achieves ~33% accuracy, while commercial systems like GLM-4.5 and xLAM-2-70B-fc-r can reach 65.62 and 75.38, respectively on the same benchmark.
This suggests that the absolute performance ceiling of ToolWeave-generated data remains relatively low, which raises concerns about its practical impact despite the relative gains over baselines.

**Questions:**

- Have the authors attempted using a frontier or commercial LLM (e.g., GPT-5, Claude 4) within the ToolWeave pipeline to generate data, and then fine-tuned a smaller model on it for comparison?
This would clarify whether ToolWeave’s design is essential.

- Since ToolWeave explicitly relies on non-frontier LLMs, data quality is a critical issue.
Although Section 4.1 presents an API quality and coverage analysis, there seems to be no explicit quality validation of the generated dialogues themselves.
For example, did the authors conduct: Cross-validation by regenerating the same dialogue multiple times and keeping only consistent outputs?
These steps would substantially increase confidence in the dataset’s reliability.

---

> ### Author Response · Authors · 2025-11-23
> **Response to Reviewer Kjdb (1/3)**
>
> We thank the reviewer for appreciating ToolWeave's modular design, the practical use of Wikipedia and Wikidata for domain grounding, and the strength of our multi-benchmark evaluation demonstrating consistent gains over competitive baselines.
>
> ---
>
> **Weakness 1:** Key terms like "Tool Graph" lack a concise explanation (nodes/edges) upon introduction in the abstract.
>
> **Response:** We appreciate the reviewer's attention to clarity. While the abstract is intended to provide a high-level summary, we agree that ensuring the accessibility of key terms early in the text is important. To address the reviewer's concern without cluttering the abstract, we have updated the paper and added a concise definition in the Introduction (`Lines 79-80`) to explicitly state that the Tool Graph consists of APIs as nodes and parameter dependencies as edges, ensuring the concept is clear before the technical methodology is presented.
>
> ---
> \
> **Weakness 2:** The paper's core motivation: to focus on non-frontier LLMs, needs stronger justification.
>
> **Response:** We respectfully but strongly disagree with the characterization of this motivation as a "false need." We argue that developing high-quality data synthesis pipelines for non-frontier, open-license models is critical for three specific reasons supported by our findings and the broader field.
>
> **1. Importance of Pipeline Architecture:** The assumption that frontier models "readily produce such data" is empirically challenged by our results. Llama-3.1-70B, when fine-tuned on the ToolFlow dataset (generated using gpt-oss-120b), achieved 21.00% on BFCL-V3 (Table 4). In contrast, the same model fine-tuned on ToolWeave data (also generated using gpt-oss-120b) achieved 33.25%.
>
> This demonstrates that methodology (pipeline structure) is more critical than the raw intelligence of the generator. Relying solely on frontier models without the fine-grained scaffolding we propose leads to suboptimal data (e.g., lack of true multi-step dependencies), as seen in the ToolDial baseline.
>
> **2. Licensing and Commercial Usability:** Existing pipelines face two legal bottlenecks that ToolWeave eliminates:
>
> - **Model Licensing:** Frontier models (e.g., GPT-4) often have Terms of Use prohibiting the use of outputs to train competing models. ToolWeave provides a "recipe" using open-weights models, ensuring the resulting data is free from such distillation restrictions.
> - **API Licensing:** As noted in our Related Work, prior methods rely on real-world APIs (e.g., RapidAPI), which introduce third-party licensing constraints. To the best of our knowledge, ToolWeave is the first framework to **synthesize APIs for complete domains from publicly available knowledge sources**, granting users complete ownership over both the dialogue data and the underlying tool schemas.
>
> **3. Reproducibility and Transparency:** Finally, while many prior works do not release their code or data, ToolWeave offers a stable, transparent solution. Because we rely on open weights rather than non-stationary commercial APIs, our pipeline is fully reproducible. This empowers the community to synthesize data for specific domains locally without ever exposing their data to third-party providers: a privacy constraint that other pipelines do not satisfy.

---

> ### Author Response · Authors · 2025-11-23
> **Response to Reviewer Kjdb (2/3)**
>
> **Weakness 3:** Evaluation ceiling and benchmark interpretation: Absolute accuracy remains low (~33% on BFCL-V3), suggesting limited practical impact compared to commercial systems (65-75%).
>
> **Response:** We thank the reviewer for raising this concern. We address it from two complementary angles: (1) generalization across diverse benchmarks, and (2) proper interpretation of BFCL scores in light of data sources, model scale, and known overfitting phenomena.
>
> **1. Broader generalization beyond BFCL:**
> We extended our evaluation to compare ToolWeave-tuned models directly against the **xLAM-2** family (specifically **xLAM-2-70b-fc-r**) on three additional benchmarks: API Bank L1, API Bank L2, CONFETTI, and ToolHop. As shown in the table below, our **Llama-3.1-70B + ToolWeave** model outperforms the specialized **xLAM-2-70B** on 3 out of 4 benchmarks (API Bank L1, API Bank L2, and CONFETTI). This indicates that ToolWeave-generated data yields robust, generalizable capabilities rather than narrow optimization for a single leaderboard.
>
> \\[
> \\begin{array}{|l|c|c|c|c|}
> \\hline
> \\textbf{Model} & \\textbf{API Bank L1} & \\textbf{API Bank L2} & \\textbf{Confetti} & \\textbf{ToolHop} \\\\
> \\hline
> \\mathrm{Llama\\text{-}3.1\\text{-}8B\\text{-}Instruct} & 62.91 & 59.46 & 20.55 & 10.05 \\\\
> \\mathrm{+\\ FT\\ on\\ ToolDial\\ (GPT\\text{-}4o\\text{-}seed)} & 59.15 & 52.70 & 23.12 & 9.55 \\\\
> \\mathrm{+\\ FT\\ on\\ ToolFlow\\ (GPT\\text{-}OSS\\text{-}seed)} & 67.67 & 55.41 & 17.19 & 16.98 \\\\
> \\mathbf{\\mathrm{+\\ FT\\ on\\ ToolWeave\\ (GPT\\text{-}OSS\\text{-}seed)}} & \\mathbf{68.92} & \\mathbf{60.81} & \\mathbf{36.76} & 21.61 \\\\
> \\mathrm{xLAM\\text{-}2\\text{-}8b\\text{-}fc\\text{-}r} & 67.17 & 58.11 & 26.88 & \\mathbf{30.05} \\\\
> \\hline
> \\mathrm{Llama\\text{-}3.1\\text{-}70B\\text{-}Instruct} & 54.89 & 59.46 & 33.00 & 11.46 \\\\
> \\mathrm{+\\ FT\\ on\\ ToolDial\\ (GPT\\text{-}4o\\text{-}seed)} & \\mathbf{71.93} & 40.54 & 11.46 & 5.83 \\\\
> \\mathrm{+\\ FT\\ on\\ ToolFlow\\ (GPT\\text{-}OSS\\text{-}seed)} & 65.66 & 60.81 & 21.34 & 12.76 \\\\
> \\mathbf{\\mathrm{+\\ FT\\ on\\ ToolWeave\\ (GPT\\text{-}OSS\\text{-}seed)}} & 71.18 & \\mathbf{64.86} & \\mathbf{45.45} & 22.51 \\\\
> \\mathrm{xLAM\\text{-}2\\text{-}70b\\text{-}fc\\text{-}r} & 65.66 & 55.41 & 31.82 & \\mathbf{36.68} \\\\
> \\hline
> \\end{array}
> \\]
>
> **2. Interpreting High BFCL Scores and Possible Overfitting:** While xLAM models achieve high scores on BFCL, this may not reflect broad generalization. The BFCL authors themselves (Patil, Shishir G., et al. "The Berkeley Function Calling Leaderboard (BFCL): From Tool Use to Agentic Evaluation of Large Language Models." ICML 2025), in Section 5.5, highlight substantial overfitting concerns with the xLAM family of models. They report that **"in contrast, Salesforce xLAM-7B (Zhang et al., 2024) shows an increase in perplexity from 3.67 to 5.09 (char-NLL from 0.340 to 0.427) on crowd-sourced, representing a notable performance degradation,"** which possibly indicates the model **"has primarily memorized benchmark solutions or was overly tuned to the static test distribution."**
>
> In contrast, our ToolWeave models show consistent gains across four distinct benchmarks, confirming that our synthetic data pipeline improves generic tool-calling capability.
>
> > **Reference:** Patil, Shishir G., et al. "The Berkeley Function Calling Leaderboard (BFCL): From Tool Use to Agentic Evaluation of Large Language Models." Forty-second International Conference on Machine Learning.
>
> **Furthermore:**
> - **xLAM-2 models are trained on GPT-4-generated data,** which is *not* license-permissible for commercial fine-tuning, and their data generation pipeline is not publicly available. ToolWeave, in contrast, uses only open-weight models and releases its full generation pipeline, ensuring reproducibility and commercial usability.
> - **GLM-4.5 (355B parameters)** is not in the comparable size class for our evaluation ($\\leq 70B$). Its much higher BFCL score reflects a different operating regime rather than a limitation of ToolWeave-generated data.

---

> > ### Author Response · Authors · 2025-11-23
> > **Response to Reviewer Kjdb (3/3)**
> >
> > **Question 1:** Have the authors attempted using a frontier or commercial LLM (e.g., GPT-5, Claude 4) within the ToolWeave pipeline to generate data, and then fine-tuned a smaller model on it for comparison? This would clarify whether ToolWeave's design is essential.
> >
> > **Response:** We thank the reviewer for this suggestion. We effectively performed this comparison in the paper (Table 4) by evaluating mistral-medium-2505 (a proprietary, frontier-class commercial model) alongside the open gpt-oss-120b. The results were illuminating: Llama-3.1-70B fine-tuned on data from the open model achieved **33.25%** accuracy, actually outperforming the version trained on the commercial model seed (**28.88%**).
> >
> > This empirical evidence clarifies two key points:
> >
> > **1. Design is Essential:** The fact that the open model matches (or beats) the commercial model proves that **ToolWeave's fine-grained scaffolding is the primary driver of data quality**, effectively bridging the capability gap between open and frontier models.
> >
> > **2. Diminishing Returns of Frontier Models:** ToolWeave is engineered specifically to mitigate the weaknesses of non-frontier models by breaking synthesis into small, verifiable steps. As a result, stronger LLMs provide diminishing returns within our architecture. Because ToolWeave already supplies the reasoning structure that frontier models typically provide, we do not expect GPT-5 to meaningfully change the conclusions. While using GPT-5 may slightly improve fluency or lexical variation, we do not expect a qualitative jump in data quality, because the pipeline already controls the reasoning and structure. Furthermore, generating our full dataset using commercial frontier models is estimated to cost $\\$$400 (GPT-5-mini) to $\\$$1000 (GPT-5) per iteration, creating a prohibitive barrier for reproducible open research compared to using open-weights models.
> >
> > That said, we are currently trying to arrange resources to extend our evaluation to GPT-5. Should these experiments be completed within the rebuttal period, we will report them in a future revision.
> >
> > ---
> > \
> > **Question 2:** Explicit quality validation of the generated dialogues is missing, which is critical for non-frontier LLMs; Employing cross-validation by regenerating the same dialogue multiple times to increase dataset reliability by retaining consistent outputs.
> >
> > **Response:** We thank the reviewer for raising the critical issue of data quality when relying on non-frontier LLMs. We appreciate the suggestion regarding cross-validation; however, we would like to clarify that dialogue quality validation is central to our analysis, and consistency is enforced architecturally rather than through post-hoc filtering.
> >
> > **1. Extensive Dialogue Quality Analysis in Sections 4.2 and 4.3:** The reviewer noted the API analysis in Section 4.1 and queried the existence of dialogue validation. We respectfully direct the reviewer's attention to the immediately following sections, Section 4.2 and Section 4.3, which analyze both the structural and semantic quality of the generated dialogues.
> >
> > - **Structural Complexity (Section 4.2):** We quantify quality using "True multi-step turns" (dependencies where tool output $A$ feeds tool input $B$ in the same turn). As shown in our structural analysis, ToolWeave (GPT-OSS) achieves **31.69%** True multi-step turns, whereas baselines like ToolFlow and ToolDial drop to **0.0%**.
> > - **Semantic Quality (Section 4.3 & Table 3):** Following ToolFlow, we employ a rigorous **LLM-as-a-judge protocol** (using the frontier-class Llama-3-405B) to evaluate dialogues on Naturalness, Coherence, Helpfulness, and Accuracy. As detailed in **Table 3**, ToolWeave consistently outperforms baselines across all dimensions (e.g., Coherence score of **4.96** vs. **4.68** for ToolDial).
> >
> > **2. Consistency by Design:** We clarify that post-hoc cross-validation is redundant because our architecture enforces consistency *a priori*. The **Fine-Grained Planner** locks the entire dialogue trajectory before the actual dialogue synthesis begins. The dialogue generator's role is strictly limited to "filling in" entities and surface wording within this fixed skeleton, ensuring consistency without the need for sampling-based filtering.
> >
> > This "construction-first" approach provides higher reliability by design than the sampling-and-filtering methods typically required for unguided generation.

---

> > > ### Comment · Reviewer_Kjdb · 2025-11-27
> > >
> > > Thank you for the clarifications. While I acknowledge the improved results on API Bank, this benchmark is relatively dated and no longer a strong indicator of current LLM function-calling capabilities. The fact that xLAM was not evaluated on API Bank yet performs better on BFCL further suggests that your approach still has room to improve in terms of broader generalization. Therefore, I am maintaining my original score.

---

> > > > ### Author Response · Authors · 2025-11-27
> > > > **Response to Reviewer Kjdb's comment (1/2)**
> > > >
> > > > We thank the reviewer for the continued engagement. We respectfully offer a counterpoint regarding generalization, specifically highlighting CONFETTI, and present new results using GPT-5-Mini as the data-generating LLM inside the ToolWeave pipeline.
> > > >
> > > > ---
> > > >
> > > > ### **1. Generalization Across Benchmarks**
> > > > - **Note on API-Bank and CONFETTI:**
> > > >   - While we acknowledge that API-Bank is a slightly older benchmark, it remains widely used in recent works such as *ToolRL* (NeurIPS 2025)`[1]`, *Planning with Multi-Constraints* (COLING 2025)`[2]`, *CITI* (AAAI 2025)`[3]`, and *Nemotron-Research-Tool-N1* (from Nvidia Labs)`[4]`. This demonstrate that it is still considered a meaningful testbed for measuring tool-use capabilities of LLMs.
> > > >   - More importantly, our evaluation relies equally on **CONFETTI** (Alkhouli et al., ACL 2025). This is a recent benchmark that comprises of complex, **human-generated** multi-turn dialogues, on which ToolWeave improves Llama-3.1-70B to 45.45%, surpassing xLAM-2-70B's 31.82% by 13.63 points. This demonstrates that ToolWeave's improvements are not restricted to older benchmarks.
> > > >   - Notably, the base **Llama-3.1-70B-Instruct** achieves **33.00%** on CONFETTI, meaning that **xLAM-2-70B (31.82%) actually degrades performance relative to its base model**, further indicating overfitting to its specific training distribution.
> > > >
> > > > - **High BFCL score does not correspond to generalization:**
> > > >   - The reviewer suggests that xLAM's higher BFCL score implies better generalization. We respectfully argue the opposite: **Generalization is defined by consistent performance across diverse distributions.**
> > > >   - **xLAM-2** models perform well on BFCL but degrades significantly on other benchmarks like CONFETTI and API Bank. This aligns with findings from the BFCL authors (Patil et al., 2025), who identified the xLAM family of models to be exhibiting signs of **overfitting** to the BFCL test set while **ToolWeave** demonstrates consistent, high performance across **BFCL, API Bank, and CONFETTI**.
> > > >
> > > > ToolWeave-tuned models exhibit consistently strong performance across BFCL, CONFETTI, API-Bank, and ToolHop, indicating broader generalization compared to the xLAM-2 model that demonstrates **narrow optimization** for a specific leaderboard.
> > > >
> > > > **References:**
> > > > 1. Qian, Cheng, et al. "ToolRL: Reward is All Tool Learning Needs." Advances in Neural Information Processing Systems, 2025.
> > > > 2. Zhang, Cong, et al. "Planning with multi-constraints via collaborative language agents." Proceedings of the 31st International Conference on Computational Linguistics, 2025.
> > > > 3. Hao, Yupu, et al. "CITI: Enhancing Tool Utilizing Ability in Large Language Models Without Sacrificing General Performance." Proceedings of the AAAI Conference on Artificial Intelligence, 2025.
> > > > 4. Zhang, Shaokun, et al. "Nemotron-Research-Tool-N1: Exploring Tool-Using Language Models with Reinforced Reasoning." arXiv preprint arXiv:2505.00024 (2025).

---

> ### Author Response · Authors · 2025-11-27
> **Response to Reviewer Kjdb's comment (2/2)**
>
> ### **2. Frontier Model Ablation: GPT-5-Mini**
> As suggested by the reviewer, we integrated **GPT-5-Mini** into the ToolWeave pipeline to empirically test the necessity of our design when using frontier models. *Note:* All the prompts and hyperparameters are fixed throughout the synthesis LLM ablations to ensure a fair comparison.
>
> The results in the table below show a modest improvement (**+2.00% overall**), validating our aforementioned hypothesis that stronger LLMs provide **diminishing returns** within our architecture that is built around the capabilities of non-frontier LLMs. Because ToolWeave already supplies the deterministic reasoning structure (planning, state tracking), GPT-5-mini primarily contributes fluency rather than a qualitative jump in logic.
>
> \\[
> \\begin{array}{|l|c|c|c|c|c|}
> \\hline
> \\textbf{Setting} & \\textbf{Multi-Turn Acc} & \\textbf{Base} & \\textbf{MissFunc} & \\textbf{MissParam} & \\textbf{LongCtx} \\\\
> \\hline
> \\text{ToolWeave (GPT-OSS-seed)} & 19.88 & 23.00 & \\mathbf{21.50} & 15.50 & 19.50 \\\\
> \\hline
> \\text{ToolWeave (GPT-5-mini-seed)} & \\mathbf{21.88} & 23.00 & 21.00 & \\mathbf{24.00} & 19.50 \\\\
> \\hline
> \\end{array}
> \\]
>
> **Complexity of GPT-5-mini generated data:** GPT-5-mini synthesized significantly richer schemas (Avg. **5.01 parameters/tool** vs 3.60). Because our training examples include the full definitions of all available tools in the context, these richer schemas resulted in much **longer contexts averaging ~20.9k tokens/sample** (vs ~11.6k).
>
> We maintained a standardized 16k training context window across all experiments to ensure a controlled comparison within a fixed compute budget. Due to this constraint, **40% of the GPT-5-mini generated examples were dropped entirely** (vs. negligible loss for GPT-OSS). This results in a performance lower bound because the model was trained on only a fraction of the available high-quality data. We are currently optimizing context length (e.g., pruning unused tools) which we expect will further improve performance. We are also extending these experiments to **Llama-3.1-70B-Instruct**.
>
> **Benefits of ToolWeave:** This experiment reinforces the advantages of our non-frontier approach:
> * **Prohibitive Cost:** Generating this iteration of dataset generation with GPT-5-mini cost us nearly **$400**. Scaling this to the bigger GPT-5 model would increase costs by ~5x, creating a barrier for reproducible research.
> * **Commercial Viability:** Models like xLAM-2 rely on GPT-4/5 data, which often prohibits commercial deployment. ToolWeave enables the creation of tool calling data using **open-source non-frontier models** using entirely **license-friendly** workflows.
> * **Data Sovereignty:** Our approach allows users to **synthesize sensitive domains locally**, resolving privacy constraints associated with exposing proprietary schemas to third-party providers.
>
> We hope these clarifications address the reviewer's concerns. We again thank the reviewer for the valuable time and feedback.

---

> > ### Author Response · Authors · 2025-11-30
> > **Updated results for frontier model ablation (GPT-5-mini)**
> >
> > As promised, we have completed the context optimization experiments for **GPT-5-mini**. We also include our existing **Mistral-Medium (Commercial)** baseline from the paper and a new **Mistral-Small-24B (Open)** ablation we did during rebuttal to provide a comprehensive comparison across model scales and licensing types.
> >
> > **1. Experimental Setup:**
> > In our original training setup, we passed the schemas of all tools in the domain to the model's context, which caused 60% of samples to exceed the 16k token limit and be dropped. To resolve this, we modified the training setup to include only the tools used in the conversation and eight distractor tools from the same domain. This ensured that 100% of the GPT-5-mini generated examples fit within the 16k token window.
> >
> > **2. Updated Empirical Results:**
> >
> > We fine-tuned Llama-3.1-8B and 70B on data generated by four different synthesis models.
> >
> > \\[
> > \\begin{array}{|c|c|c|c|c|c|}
> > \\hline
> > \\textbf{8B Model Results} & \\textbf{Acc} & \\textbf{Base} & \\textbf{MissFunc} & \\textbf{MissParam} & \\textbf{LongCtx} \\\\
> > \\hline
> > \\text{ToolWeave (Mistral-Small-24B) [Open]} & 20.00 & 30.00 & 14.00 & 15.50 & 20.50 \\\\
> > \\hline
> > \\text{ToolWeave (GPT-OSS-120B) [Open]} & 19.88 & 23.00 & 21.50 & 15.50 & 19.50 \\\\
> > \\hline
> > \\text{ToolWeave (Mistral-Medium) [Commercial]} & \\mathbf{25.00} & \\mathbf{31.50} & 20.50 & \\mathbf{24.50} & \\mathbf{23.50} \\\\
> > \\hline
> > \\text{ToolWeave (GPT-5-mini) [Frontier]} & 23.12 & 25.00 & \\mathbf{22.50} & 24.00 & 21.00 \\\\
> > \\hline
> > \end{array}
> > \\]
> >
> > \\[
> > \\begin{array}{|c|c|c|c|c|c|}
> > \\hline
> > \\textbf{70B Model Results} & \\textbf{Acc} & \\textbf{Base} & \\textbf{MissFunc} & \\textbf{MissParam} & \\textbf{LongCtx} \\\\
> > \\hline
> > \\text{ToolWeave (Mistral-Small-24B) [Open]} & 28.00 & 29.00 & 32.50 & 26.50 & 24.00 \\\\
> > \\hline
> > \\text{ToolWeave (GPT-OSS-120B) [Open]} & 33.25 & 37.50 & 35.00 & \\mathbf{32.00} & 28.50 \\\\
> > \\hline
> > \\text{ToolWeave (Mistral-Medium) [Commercial]} & 28.88 & 27.50 & \\mathbf{39.00} & 30.00 & 19.00 \\\\
> > \\hline
> > \\text{ToolWeave (GPT-5-mini) [Frontier]} & \\mathbf{34.62} & \\mathbf{38.00} & 37.50 & 31.50 & \\mathbf{31.50} \\\\
> > \\hline
> > \\end{array}
> > \\]
> >
> > **3. Key Findings from these experiments:**
> > * **Architecture Compensates for Scale:** On the larger 70B model, our Open Source seed (gpt-oss-120b) outperforms the Commercial seed (mistral-medium) by +4.37% and trails the best Frontier model (gpt-5-mini) by only 1.37%. Notably, the small open-weight Mistral-Small-24B (28.00%) performs on par with the commercial Mistral-Medium (28.88%). This proves that ToolWeave's fine-grained scaffolding is the primary driver of data quality.
> >
> > * **Cost-Benefit Analysis:** Achieving this marginal gain required using a commercial API that costs significantly more per iteration compared to open-weights inference. This underscores the value of ToolWeave in enabling the community to generate SOTA-level, license-friendly data using accessible non-frontier models.

---

### Official Review · Reviewer_KXxZ · 2025-10-31

**Soundness:** 2
**Presentation:** 3
**Contribution:** 3
**Rating:** 6
**Confidence:** 3

**Summary:**

This paper introduces ToolWeave, a fully open-source and fine-grained controllable framework for synthesizing multi-turn tool-calling datasets using non-frontier large language models. Unlike prior pipelines that rely on proprietary models such as GPT-4, ToolWeave builds everything from scratch with open knowledge sources (Wikipedia, Wikidata) and a modular five-stage process: tool graph synthesis, structured sampling, fine-grained plan generation, multi-agent dialogue execution, and post-processing. This design enables weaker open models to produce coherent, realistic, and robust tool-use dialogues.

**Strengths:**

Structured and controllable generation pipeline:
The framework employs a five-stage modular design—from tool graph synthesis to dialogue post-processing—allowing precise control over each step, making the generation process transparent, reproducible, and easy to debug.

Reliance on open-source models:
ToolWeave operates entirely with non-frontier, open-source LLMs and open knowledge bases, ensuring license-friendly data generation without dependence on proprietary systems.

Significant performance improvement and strong generalization:
Models fine-tuned on ToolWeave data achieve substantial gains and robust generalization across multiple benchmarks, outperforming prior pipelines such as ToolFlow and ToolDial.

**Weaknesses:**

1. The paper appears to lack a clear description of any human verification process. While the goal of developing a fully automated pipeline is understandable, incorporating at least light sampling or manual validation would strengthen reliability. For instance, in the initial stage where the model generates APIs, it is unclear how the authors ensure that these APIs are logically consistent and error-free, as issues such as inconsistent parameter types or missing logical dependencies may still occur.

2. The synthesized APIs may not fully reflect the complexity of real-world APIs. Since they lack real execution, latency, and side effects, there could be a gap between the synthetic and practical tool-calling scenarios.

3. In several benchmarks, ToolDial and ToolFlow appear to negatively affect model performance after SFT. It would be helpful if the authors could provide a finer-grained analysis to understand why this happens—perhaps through ablations or error case studies.

**Questions:**

1. It remains somewhat unclear whether being “fully open-source and license-friendly” alone constitutes a substantial contribution. Since many existing tool-calling datasets can already be generated locally using open models within sandboxed environments, the paper should clarify more explicitly what distinguishes ToolWeave in this aspect.

---

> ### Author Response · Authors · 2025-11-23
> **Response to Reviewer KXxZ (1/2)**
>
> We appreciate the reviewer's recognition of ToolWeave's structured and controllable multi-stage pipeline, its reliance on open-source models, and its strong performance gains across diverse benchmarks.
>
> ---
>
> **Weakness 1:** No human verification process is described to ensure the logical consistency and error-free nature of the generated APIs.
>
> **Response:** We appreciate the reviewer's suggestion regarding reliability. While we agree that human-in-the-loop verification can strengthen dataset quality, our primary research objective is to enable the scalable generation of high-quality, license-friendly multi-turn dialogues without human intervention. To ensure reliability without manual bottlenecks, our pipeline substitutes human oversight with fine-grained structural control and automated validation mechanisms.
>
> To specifically address concerns about logical consistency and parameter errors in the API synthesis stage without human intervention, we rely on three core design mechanisms that inherently prevent these issues:
>
> - **Knowledge Grounding:** By grounding schemas in Wikipedia and Wikidata, the synthesized APIs inherit real-world vocabulary and relationships, ensuring logical consistency across the generated tools.
> - **Curriculum-Driven Synthesis:** We avoid the errors common in "one-shot" generation by decomposing API creation into verifiable stages: Seed Generation $\\rightarrow$ Entity Expansion $\\rightarrow$ Schema Enrichment. This iterative process allows us to validate APIs incrementally.
> - **Automated Structural Validation:** As detailed in the paper, proposed edges in the Tool Graph are explicitly "validated by an LLM to confirm that data flows are semantically correct". Every stage of the API synthesis includes automated syntactic and embedding-based checks to prune inconsistent schemas.
>
> These automated mechanisms produced APIs and dialogues that achieve strong empirical performance across benchmarks, demonstrating that full automation is feasible for non-frontier LLMs. **At the same time, ToolWeave's modular design allows human intervention at any stage, enabling users to incorporate manual verification whenever higher quality assurance is desired.**
>
> ---
> \
> **Weakness 2:** Synthesized APIs are simplistic, lacking real-world characteristics like execution, latency, and side effects.
>
> **Response:** We appreciate the reviewer's observation regarding the gap between synthetic and real-world environments. Below we clarify how ToolWeave addresses these issues.
>
> **1. Comparison with Real-World APIs :** Contrary to the concern that synthetic APIs lack complexity of real-world APIs, our analysis (Section 4.1) confirms that our generated schemas often exceed the structural complexity of standard real-world APIs. As shown below, ToolWeave generates APIs with significantly higher parameter density and nesting compared to 4,474 real-world APIs from RapidAPI:
>
> \\[
> \\begin{array}{|c|c|c|c|}
> \\hline
> \\textbf{Metric} & \\textbf{Real-World APIs (RapidAPI)} & \\textbf{ToolWeave (GPT-OSS)} & \\textbf{ToolWeave (Mistral Medium)} \\\\
> \\hline
> \\text{Avg. Params / API} & 2.4 & 3.60 & 2.92 \\\\
> \\hline
> \\text{Complex API Use (CAU)} & 1.0\\% & 23.6\\% & 19.8\\% \\\\
> \\hline
> \\text{Interconnectivity} & 1.61^* & 1.08 & 1.85 \\\\
> \\hline
> \\end{array}
> \\]
>
> > *As discussed in the paper, the RapidAPI score is inflated by false positives from generic parameter matching (e.g., "id"), whereas our graph is causally validated.
>
> These metrics indicate that our curriculum-driven synthesis produces tools with high parameter density and deep nesting, effectively mitigating the risk of over-simplification.
>
> **2. Simulating Side Effects via Robustness Injection:** While we do not use live execution, we explicitly simulate its consequences, specifically failures and side effects, via our **Dialogue Post-Processing** stage (Section 3.5). We deterministically inject realistic artifacts such as Error Recovery (simulating API failures) and Missing-Function scenarios. This trains the model fine-tuned on ToolWeave generated data to handle the logical fallout of execution (e.g., missing dependencies or refusal).
>
> **3. Latency:** We firmly clarify that latency is strictly an inference-time system metric, irrelevant to the model's reasoning accuracy. The semantic correctness of a tool call is independent of execution speed; therefore, latency impacts neither the semantic validity nor the reasoning complexity of the training data required for fine-tuning.

---

> > ### Author Response · Authors · 2025-11-23
> > **Response to Reviewer KXxZ (2/2)**
> >
> > **Weakness 3:** A finer-grained analysis is needed to explain why ToolDial and ToolFlow negatively affect SFT performance.
> >
> > **Response:** We thank the reviewer for highlighting the need to understand why ToolDial and ToolFlow can degrade model performance after SFT. We conducted a fine-grained analysis combining structural metrics (Section 4.2) and qualitative error analysis (Appendix J & K) to diagnose the underlying causes.
> >
> > **1. Structural deficiencies - lack of tool dependencies:**
> > Our quantitative analysis shows that the primary issue is the absence of true multi-step reasoning in these datasets. ToolWeave contains up to **36.14%** true multi-step turns (where outputs from one tool feed directly into another in the same turn), while **ToolFlow and ToolDial exhibit 0.0%**. As a result, models trained on these baselines never observe dependent tool chains and therefore fail to generalize to multi-step workflows at evaluation time. Although ToolFlow produces long dialogues (~69 turns), this is due to its high-level plans fail to guide non-frontier LLMs on how to progress through the task. As a result, the model often enters uncontrolled loops of independent tool calls rather than executing a coherent, goal-directed workflow.
> >
> > **2. Qualitative issues - hallucinations and unnatural interaction patterns:**
> > Our error analysis further reveals semantic flaws that introduce noise during SFT.
> >
> > - *ToolFlow:* **As shown in Appendix J**, tool agent hallucinates a tool response without the assistant agent providing an actual tool call, resulting in a fabricated tool reply.
> > - *ToolDial:* **As shown in Appendix K**, dialogues often lack proper error-recovery turns and exhibit unrealistic user behavior. Assistants frequently fail to request missing parameters (e.g., the hotel IP address in Example K.1) and users speak in ways that do not reflect natural interaction patterns (e.g., using backend-style parameter names or unnatural request formulations in Example K.2), creating a distribution shift away from real-world dialogues.
> >
> > So, the negative transfer arises from a combination of shallow reasoning structure and low-quality interaction patterns. These deficiencies lead LLMs to learn brittle or incorrect behaviors during SFT, explaining why ToolFlow and ToolDial can harm downstream tool-calling performance.
> >
> > ---
> > \
> > **Question 1:** The paper needs to clarify what substantially distinguishes its "fully open-source" contribution from existing locally generatable datasets.
> >
> > **Response:** We appreciate the reviewer's question to clarify the specific contribution of our "fully open-source" approach. Generating *high-quality, commercially usable* data using only non-frontier models is a problem that prior work has not solved. ToolWeave distinguishes itself by addressing three critical barriers:
> >
> > **1. Importance of Pipeline Architecture**
> >
> > The assumption that open models can readily produce high-quality data is challenged by our results. Without ToolWeave's specific scaffolding, open models struggle to generate coherent multi-step reasoning. This is proven by our comparison in Table 4 of the main paper:
> >
> > - **ToolFlow Data (generated by gpt-oss-120b):** Fine-tuning Llama-3.1-70B yields 21.00% on BFCL-V3.
> > - **ToolWeave Data (generated by gpt-oss-120b):** Fine-tuning Llama-3.1-70B yields 33.25% on BFCL-V3.
> >
> > This demonstrates that pipeline structure is a critical determinant of data quality, capable of compensating for differences in raw generator intelligence.
> >
> > **2. Licensing and Commercial Usability**
> >
> > Existing pipelines face two legal bottlenecks that ToolWeave eliminates:
> >
> > - **Model Licensing:** Frontier models (e.g., GPT-4) often have Terms of Use prohibiting the use of outputs to train competing models. ToolWeave provides a "recipe" using open-weights models, ensuring the resulting data is free from such distillation restrictions.
> > - **API Licensing:** As noted in our Related Work, prior methods rely on real-world APIs (e.g., RapidAPI), which introduce third-party licensing constraints. To the best of our knowledge, ToolWeave is the first framework to **synthesize APIs for complete domains from publicly available knowledge sources**, granting users complete ownership over both the dialogue data and the underlying tool schemas.
> >
> > **3. Reproducibility and Transparency:** Finally, while many prior works do not release their code or data, ToolWeave offers a stable, transparent solution. Because we rely on open weights rather than non-stationary commercial APIs, our pipeline is fully reproducible. This empowers the community to synthesize data for specific domains locally without ever exposing their data to third-party providers, a privacy constraint that other pipelines do not satisfy.

---

> > > ### Author Response · Authors · 2025-11-27
> > > **Follow-up on rebuttal responses**
> > >
> > > Dear Reviewer KXxZ,
> > >
> > > We sincerely appreciate the time and effort you have dedicated to reviewing our work.  We have responded thoroughly to your main concerns in our previous responses. As the end of the discussion period is approaching, could you kindly let us know if there are any outstanding issues or further clarifications needed so that we can address them in a timely manner? Your feedback has been invaluable, and we appreciate the opportunity to refine our work.

---

### Official Review · Reviewer_WxXD · 2025-11-01

**Soundness:** 2
**Presentation:** 2
**Contribution:** 2
**Rating:** 4
**Confidence:** 4

**Summary:**

This paper presents ToolWeave, a modular and controllable synthetic data generation pipeline for multi-turn tool calling using non-frontier LLMs. The method constructs a domain-specific synthetic Tool Graph from open sources (Wikipedia and Wikidata), samples structured tool workflows based on common motifs (linear, fan-in/out, conditional), and generates fine-grained dialogue plans that are then instantiated through a multi-agent dialogue synthesizer. The final dialogues are post-processed to enhance robustness and diversity. Experiments on BFCL-V3, API Bank, CONFETTI, and ToolHop benchmarks show that data generated by ToolWeave can significantly improve tool-calling performance of open LLMs.

**Strengths:**

1. The paper targets an important and timely problem: generating license-friendly, high-quality multi-turn tool-calling data without relying on frontier models.
2. The paper evaluates across multiple benchmarks and base models, demonstrating consistent improvements. Results are strong and well-analyzed.
3. The authors promise to release code, tools, and data, which will be valuable to the community.

**Weaknesses:**

1. While the system is well-engineered, many ideas (e.g., tool graph construction, dialogue synthesis via plans) have been explored in ToolFlow, Magnet, and APIgen-MT. The novelty lies primarily in the granularity and control, which should be emphasized more clearly in the introduction and discussion.

2. Since the APIs are synthesized by LLMs, it is unclear how faithfully they represent real-world API semantics. And the bias in LLMs will be amplified in the continual training with the synthesized dataset.

3. Since the tool responses are role-played by LLMs rather than generated by real API executions, the responses in multi-turn or multi-step dialogues are always idealized and deterministic. In realistic scenarios, however, API responses are often unexpected, containing errors, missing fields, or inconsistent outputs that require the model to perform error correction or recovery. This mismatch introduces a systematic bias in the synthesized dataset, making the resulting fine-tuned models less robust to noisy or imperfect tool outputs.

4. The ablation study could better justify the necessity of each stage (graph sampler, fine-grained planner, post-processor).

5. Although the text describes tool motifs in detail, the paper lacks a clear visualization of the generated Tool Graph.

**Questions:**

Please refer to the weaknesses

---

> ### Author Response · Authors · 2025-11-23
> **Response to Reviewer WxXD (1/3)**
>
> We sincerely appreciate the reviewer's recognition of ToolWeave's timely focus on license-friendly data generation, the rigor of our multi-benchmark evaluation, and our commitment to releasing all resources to the community.
>
> ---
>
> **Weakness 1:** The core novelty of the work (granularity and control) is not clearly distinguished from prior efforts (ToolFlow, Magnet, APIGen-MT).
>
> **Response:** We thank the reviewer for this insightful comment and agree that clarifying our specific contributions relative to prior art is crucial. Concepts like tool graphs and plan-based synthesis have been explored. Our novelty, as you noted, lies in the granularity and control of these components, which are specifically designed to enable non-frontier LLMs to generate high-quality data.
>
> ---
>
> **Tool Graph Construction:** While prior works like ToolFlow, ToolDial, Magnet, and APIGen-MT also use tool graphs, they rely on APIs from marketplaces like RapidAPI. Consequently, their graph construction relies on post-hoc, heuristic edge inference, such as using semantic similarity between the names and descriptions of the parameters of these APIs. As we note in our analysis (Section 4.1), this method can introduce "false positives" (e.g., linking generic 'id' parameters) or ambiguous "soft" matches.
>
> Our novelties, in contrast, are:
>
> a) The Tool Graph Synthesizer is prompted to generate tools "with candidate links in mind" (e.g., `search_hotel` produces a `hotel_id` for `book_hotel` to consume).
>
> b) These proposed parameter-level connections are then explicitly "validated by an LLM to confirm that data flows are semantically correct".
>
> c) This process results in a "cleaner, causally-sound Tool Graph" built on explicit, validated data flows, rather than a graph based on semantic similarity that introduces false edges between tools.
>
> ---
>
> **Dialogue Plan Granularity:** Similarly, while frameworks like ToolFlow, APIGen-MT use plans, our "Fine-Grained Plan Generator" provides a level of explicit scaffolding designed specifically to address the failure modes of non-frontier LLMs, such as hallucinated arguments and state loss.
>
> Our JSON plan is not just a high-level goal sequence; it is a deterministic, step-by-step specification that encodes:
>
> a) The acting agent at each turn (User, Assistant, Tool).
>
> b) The specific subgoal being advanced.
>
> c) Crucially, how every tool parameter is sourced (e.g., from the user, an upstream tool output, or a schema default).
>
> d) Where to inject clarification or error turns deterministically.
>
> This contrasts with baseline methods, which from our analysis (Section 4.2) found to produce "uncontrolled" dialogues (ToolFlow) or "shallow" dialogues with 0.0% "True multi-step" turns (ToolDial). **"True Multi-step"** turns are the turns where "tool calls are directly dependent, meaning the output of one tool call is consumed by another within the same turn." These turns serve as a rigorous proxy for the planning capability of tooling LLM.
>
> We agree that granularity and control are the central features that distinguish our contribution. We have revised the introduction to better emphasize that this fine-grained, validated, and deterministic "scaffolding" is our primary novelty and is the key architectural choice that enables non-frontier LLMs to succeed at this complex multi-turn tool calling data synthesis task. Furthermore, our experimental results, both in the main paper and in the ablation study provided in response to Weakness 4, empirically support our hypothesis that an LLM trained on dialogues generated from these fine-grained plans achieves superior benchmark performance. This confirms that such granularity is a prerequisite for non-frontier LLMs.

---

> > ### Author Response · Authors · 2025-11-23
> > **Response to Reviewer WxXD (2/3)**
> >
> > **Weakness 2:**  Since the APIs are synthesized by LLMs, it is unclear how faithfully they represent real-world API semantics. And the bias in LLMs will be amplified in the continual training with the synthesized dataset.
> >
> > **Response:** We appreciate the reviewer's insight. In the paper, we have provided two lines of evidence demonstrating that our synthetic APIs are faithful to the complexity and structure of real-world APIs and are effective for fine-tuning, despite their generative nature.
> >
> > **1. Ensuring API Faithfulness:** To ensure our synthetic APIs are not simplistic or unrepresentative, we quantitatively compared them to 4,474 real-world APIs from RapidAPI. Our analysis in Section 4.1, "API Quality and Coverage," shows our APIs, which are grounded in real-world semantics from Wikipedia and Wikidata, are often more complex than their real-world counterparts.
> >
> > Specifically, our APIs feature:
> >
> > - A "Complex API Use (CAU)" score of up to **23.6%** (vs. **1.0%** for real APIs)
> > - A higher average parameter count (~**3.6** vs. **2.4**)
> >
> > This suggests they are a faithful-enough representation of real-world complexity.
> >
> > **2. Validating Data Effectiveness:** The concern about bias is best addressed empirically. The strong and consistent performance gains our data provides across four diverse benchmarks (BFCL-V3, API Bank, etc.), detailed in Section 5, "Experiments," demonstrate that models fine-tuned on our generated data are learning robust, generalizable tool-calling skills, **not** simply overfitting to the synthesis LLMs biases.
> >
> > ---
> > \
> > **Weakness 3:** The reliance on LLM role-played tool responses creates an idealized, deterministic dataset, lacking the noise, errors, and inconsistencies of real-world API outputs.
> >
> > **Response:** We thank the reviewer for this critical observation. We entirely agree that fine-tuning on purely "idealized" data would create a brittle model. This is precisely why we designed the Dialogue Post-Processing stage (Section 3.5). We have updated the paper to include a new Appendix F.1 (referenced in Sec 3.5), which details the formal algorithms for the Dialogue Error Injection component. The purpose of this component is to mitigate this exact bias by injecting realistic error patterns and robustness patterns to break the idealized flow.
> >
> > Our pipeline systematically introduces a wide variety of realistic API failures, such as:
> >
> > - `MISSING_PARAMETER`
> > - `INCORRECT_TYPE`
> > - `WRONG_TOOL`
> >
> > These force the model to learn error correction and recovery.
> >
> > This robustness training is a key contributor to our model's strong performance on challenging benchmarks like BFCL-V3. As shown in Table 4 of the main paper, fine-tuning the Llama-3.1-70B model on our data resulted in large absolute gains, **+22.0%** in the "Missing Function" category, and **+21.5%** in "Missing Parameter" category, demonstrating the effectiveness of our data at teaching crucial skills.

---

> > > ### Author Response · Authors · 2025-11-23
> > > **Response to Reviewer WxXD (3/3)**
> > >
> > > **Weakness 4:** The ablation study could better justify the necessity of each stage (graph sampler, fine-grained planner, post-processor).
> > >
> > > **Response:** We thank the reviewer for this constructive suggestion. To empirically justify the necessity of each pipeline stage, specifically the Graph Sampler, Fine-Grained Planner, and Post-Processor, we conducted a comprehensive ablation study using Llama-3.1-8B-Instruct on the BFCL-V3 benchmark.
> > >
> > > ---
> > >
> > > **1. Necessity of Dialogue Post-Processing:** Table 1 isolates the impact of the post-processing stage. Injecting robustness patterns yields a substantial **+6.26%** absolute gain in Multi-Turn Accuracy. Evaluated on Llama-3.1-8B-Instruct, post-processing produces large and consistent gains across all metrics.
> > >
> > > **Table 1: Performance impact of the post-processing (postproc) stage.**
> > >
> > > \\[
> > > \\begin{array}{|c|c|c|c|c|c|}
> > > \\hline
> > > \\textbf{Setting} & \\textbf{Multi-Turn Acc} & \\textbf{Base} & \\textbf{MissFunc} & \\textbf{MissParam} & \\textbf{LongCtx} \\\\
> > > \\hline
> > > \\text{ToolWeave (GPT-OSS seed) without postproc} & 13.62 & 14.50 & 14.50 & 13.50 & 12.00 \\\\
> > > \\hline
> > > \\text{ToolWeave (GPT-OSS seed) with postproc} & \\mathbf{19.88} & \\mathbf{23.00} & \\mathbf{21.50} & \\mathbf{15.50} & \\mathbf{19.50} \\\\
> > > \\hline
> > > \\text{Absolute performance gain} & +6.26 & +8.50 & +7.00 & +2.00 & +7.50 \\\\
> > > \\hline
> > > \\end{array}
> > > \\]
> > >
> > > ---
> > >
> > > **2. Necessity of Structured Sampling and Fine-Grained Planning:** Table 2 validates the architecture of the generation pipeline itself (with post-processing disabled).
> > >
> > > - **Impact of Structured Sampler:** Replacing our motif-based sampler with a random walk (Row 2) degrades accuracy from 13.62% to 12.25%.
> > > - **Impact of Fine-Grained Planner:** Row 3 shows that replacing our planner with ToolFlow collapses performance to 7.50%, demonstrating the planner is the most critical component.
> > >
> > > **Table 2: Performance comparison across settings.**
> > > Ablations via controlled component replacement. All settings exclude post-processing for comparability. We evaluate (1) replacing the structured sampler with random sampling while keeping ToolWeave's planner and synthesizer, and (2) replacing ToolWeave's planner+synthesizer with ToolFlow given the same structured goals.
> > >
> > > \\[
> > > \\begin{array}{|c|c|c|c|c|c|}
> > > \\hline
> > > \\textbf{Setting} & \\textbf{Multi-Turn Acc} & \\textbf{Base} & \\textbf{MissFunc} & \\textbf{MissParam} & \\textbf{LongCtx} \\\\
> > > \\hline
> > > \\text{ToolWeave (no postproc)} & \\textbf{13.62} & \\textbf{14.50} & \\textbf{14.50} & \\textbf{13.50} & \\textbf{12.00} \\\\
> > > \\hline
> > > \\text{Random sampling + ToolWeave (Plan + synth)} & 12.25 & 13.00 & 14.00 & \\textbf{13.50} & 8.50 \\\\
> > > \\hline
> > > \\text{Structured sampling + goals } \\to \\text{ ToolFlow} & 7.50 & 9.50 & 7.50 & 5.00 & 8.00 \\\\
> > > \\hline
> > > \\text{ToolFlow baseline (GPT-OSS seed)} & 7.62 & 11.00 & 4.50 & 5.50 & 9.50 \\\\
> > > \\hline
> > > \\end{array}
> > > \\]
> > >
> > > ---
> > > \
> > > **Weakness 5:**  Although the text describes tool motifs in detail, the paper lacks a clear visualization of the generated Tool Graph.
> > >
> > > **Response:** We thank the reviewer for this suggestion. We agree that a visual aid significantly clarifies the theoretical descriptions of the tool motifs. To address this, we have added Appendix B.3 and included Figure 2, which illustrates a representative Tool Graph from the E-commerce domain.

---

> > > > ### Author Response · Authors · 2025-11-27
> > > > **Follow-up on rebuttal responses**
> > > >
> > > > Dear Reviewer WxXD,
> > > >
> > > > We sincerely appreciate the time and effort you have dedicated to reviewing our work.  We have responded thoroughly to your main concerns in our previous responses. As the end of the discussion period is approaching, could you kindly let us know if there are any outstanding issues or further clarifications needed so that we can address them in a timely manner? Your feedback has been invaluable, and we appreciate the opportunity to refine our work.

---

### Author Response · Authors · 2025-11-23
**Comment for all reviewers**

In response to the reviewers' feedback, we have revised the paper; newly added or modified text is highlighted in blue for clarity.

---

### Meta-Review · Area_Chair_BmXv · 2025-12-16

**Summary:**

The paper presents ToolWeave, a new synthetic data generation pipeline designed to improve multi-turn tool-calling capabilities for language models without relying on proprietary frontier models or commercial APIs. Existing synthetic data approaches for tool calling either depend on costly licensed models (e.g., GPT-4) or produce low-quality dialogues with poor diversity and structural coherence. In contrast, ToolWeave builds high-fidelity, domain-grounded multi-turn dialogues using non-frontier, license-friendly LLMs by introducing three key innovations. First, it synthesizes a Tool Graph from publicly available domain text (e.g., Wikipedia, Wikidata) to create realistic and interconnected API schemas. Second, it uses a two-stage planning process that first generates coherent goals from subgraphs of the Tool Graph and then decomposes these goals into fine-grained step-by-step plans. Third, a controlled multi-agent dialogue synthesizer executes these plans with explicit state tracking and post-processing to add diversity and error patterns. Experimental results show that models fine-tuned on ToolWeave data significantly outperform baseline pipelines (e.g., ToolFlow, ToolDial) across several multi-turn tool-calling benchmarks, validating ToolWeave’s effectiveness.

**Reviewer Concerns:**

- Concern 1: the novelty of the proposed system: many ideas (e.g., tool graph construction, dialogue synthesis via plans) have been explored in ToolFlow, Magnet, and APIgen-MT.
- Concern 2: The paper’s core motivation, to focus on non-frontier LLMs, needs stronger justification. If frontier models can readily produce such data, why is it necessary to design a specialized workflow for weaker models? The motivation behind this work may be a false need.
- Concern 3: Detailed ablation study to each component in the proposed workflow.
- Concern 4: The gap between the synthetic and practical tool-calling scenarios.

I think in the rebuttal phase, the authors spent great efforts in responding to all of these concerns. I think part of the concerns (mainly Concern 3 and Concern 4) are properly addressed (as acknowledged by the reviewers). While for Concern 1 and Concern 2, I can see the novelty and the motivation of the paper lies in the granularity and control of these components, which are specifically designed to enable non-frontier LLMs to generate high-quality data. However, I feel such novelty (in comparison with previous works) fails to meet the acceptance criteria of ICLR.

**Reviewer Scores:**

Reviewer's score will increase from 6, 4, 4, 4 to 6, 6 (Reviewer JXqN), 4, 4 or 6, 6 (Reviewer JXqN), 6 (Reviewer WxXD), 4.

---

### Decision · Program_Chairs · 2026-01-26

Reject